# Measurement report: Long-term real-time characterisation of the submicronic aerosol and its atmospheric dynamic in a Mediterranean coastal city: Tracking the polluted events at the Marseille-Longchamp supersite

Benjamin Chazeau[1,2], Brice Temime-Roussel[1], Grégory Gille[2], Boualem Mesbah[2], Barbara D'Anna[1], Henri Wortham[1], Nicolas Marchand[1]

[1]Aix Marseille Univ, CNRS, LCE, Marseille, France
[2]AtmoSud, Regional Network for Air Quality Monitoring of Provence-Alpes-Côte-d'Azur, Marseille, France

*Correspondence to:* Benjamin Chazeau (benjamin.chazeau@univ-amu.fr) and Nicolas Marchand (nicolas.marchand@univ-amu.fr)

**Abstract.** A supersite was recently implemented in Marseille to conduct intensive and advanced measurement studies for ambient aerosols. A Time-of-Flight Aerosol Chemical Speciation Monitor (ToF-ACSM) was deployed to investigate the chemical composition of submicronic aerosol over a 14-month period (1 February 2017 - 13 April 2018). Parallel measurements were performed with an Aethalometer, an ultrafine particle monitor and a suite of instruments to monitor regulated pollutants ($PM_{2.5}$, $PM_{10}$, $NO_x$, $O_3$ and $SO_2$). The averaged $PM_1$ chemical composition over the period was dominated by organics (49.7%) and black carbon (17.1%) while sulfate accounted for 14.6%, nitrate for 10.2%, ammonium for 7.9% and chloride for 0.5% only. Wintertime was found to be the season contributing the most to the annual $PM_1$ mass concentration (30%), followed by autumn (26%), summer (24%) and spring (20%). During this season, OA and BC concentrations were found to contribute to 32% and 31% of their annual concentrations, respectively, as a combined result of heavy urban traffic, high emissions from residential heating, open combustion of green wastes and low planetary boundary layer (PBL) height. In summer, sulfate contribution to $PM_1$ increased with an average and a maximum contribution to the $PM_1$ of 24% and 66%. This is partly due to local photochemical production from its precursor $SO_2$, locally emitted by shipping and industrial activities and advected to the city under sea breeze conditions. Results from backtrajectory cluster analysis suggest that, besides local anthropogenic activities, Mediterranean long-range transport contributes the most to the enrichment of the sulfate fraction. Another important feature of the summer season is that half of the most intense $SO_2$ peaks happen at that time of the year and are associated to higher UFPs number.

The fifteen days exceeding the target daily $PM_{2.5}$ concentration value recommended by the World Health Organization (WHO) occurred during the cold period (late autumn-early spring). These episodes contribute to an increase of 6.5% of the annual $PM_1$ concentration. Local and long-range pollution episodes could be distinguished, accounting for 40 and 60% of the exceedance days, respectively. Enhanced OA and BC concentrations, mostly originating from domestic wood burning under nocturnal

land breeze conditions were observed during local pollution episodes, while high level of oxygenated OA and inorganic nitrate were associated to medium/long-range transported particles.

In conclusion this supersite showed a high potential for the study of seasonality and pollution episodes phenomenology in Marseille over multiple geographic scales. The present paper highlights the significant contribution of regional transport of pollutants to the local air pollution that must be considered by local authorities in deploying effective PM abatement strategies.

## 1 Introduction

A major societal concern in EU countries is related to air quality as it is nowadays recognized as the first sanitary risk from environmental origin. The exposure to fine particles causes premature mortality in European cities according to the report from Environmental European Agency (EEA, 2019). Recent models predict that premature deaths would reach 380000 per year and could even increase up to 500 000 by 2050 (Lelieveld et al., 2015). In France, according to a recent study, 48000 premature deaths per year are linked to fine particles, representing the third cause of mortality behind alcohol and tobacco (Pascal et al., 2016). Regarding annual safety threshold recommendations for both $PM_{10}$ (20 µg $m^{-3}$) and $PM_{2.5}$ (10 µg $m^{-3}$) addressed by the World Health Organization (WHO, 2006), most of European countries still exceed the limit values (48 and 68% of air quality monitoring stations, respectively). In Southern France, 85% of the population leave in areas exceeding the WHO recommendations for particle matter. Even though it is well recognized that smaller particles may cause more damage to human health than $PM_{10}$ (Kreyling et al., 2006), Air Quality Standard for $PM_{2.5}$ (and $PM_1$) has still not been set up in France. Therefore, it appears important to focus on this specific size fraction and extensive studies on high polluted episodes are needed. They are directly linked to several features as chemical composition, physical properties of fine particles, proximity of anthropogenic activities and meteorological conditions. Moreover (Pandolfi et al., 2020) described that long-range sources could affect local urban pollution as more than 60% of PM in five European urban sites were from regional and continental origins.

In the last 15 years aerosol mass spectrometer (AMS) technology dedicated to the real-time analysis of the submicron aerosol chemical composition (Canagaratna et al., 2007; Jayne et al., 2000) has been widely used to investigate the emissions and transformation processes of aerosol chemical species. Unfortunately, most of the studies are limited to short-term campaign measurements, hindering our understanding of aerosol chemical composition over long temporal scales. The recently developed Aerodyne aerosol chemical speciation monitor (ACSM) is more robust and easily deployed for long-term monitoring of submicron aerosol chemical composition (Fröhlich et al., 2013; Ng et al., 2011), even if the ability of measuring particle size distribution is no longer available and the poor resolution of ACSM mass spectra does not allow high-resolution peak fitting (Timonen et al., 2016).

Studies of several month duration using AMS were previously conducted in Europe: Melpitz (3 months; Poulain et al., 2011), Puy-de-Dôme (5 months; Freney et al., 2011) and London (1 year; Young et al., 2015). In recent years ACSM studies were performed for long-term in remote/regional sites such as Jungfraujoch (14 months; Fröhlich et al., 2015), Montseny (1 year; Minguillón et al., 2015), Ispra (1 year; Bressi et al., 2016) or Aukstaitija in Lithuania (9 months; Pauraite et al., 2019). Similarly

long-term investigations were also carried out in large European cities at urban background scale in Paris (2 years; Petit et al., 2015), Zurich (1 year; Canonaco et al., 2015), Helsinki (4 months; Aurela et al., 2015), London (10 months; Reyes-Villegas et al., 2016) and Athens (1 year ; Stavroulas et al., 2019).

The Mediterranean city of Marseille, as a highly urbanised area, exposed to a variety of anthropic (traffic, residential heating, shipping, industries) and biogenic (terrestrial vegetation, marine aerosols) sources is a challenging area for fine particle studies. The ESCOMPTE experiment (2001) demonstrated that the topography and the air mass circulation, characterized by local and mesoscale winds, drives the pollution levels in the city (Cachier et al., 2005; Drobinski et al., 2007; Mestayer et al., 2005; Puygrenier et al., 2005). High level of atmospheric pollutants such as fine particles, have often been observed in Marseille, where mortality rate and cardiovascular hospital admissions are significantly elevated, even higher than in Paris whose population is 6 times higher (Pascal et al., 2013).

Fine particles have been previously characterized in Marseille during an intensive field campaign (3 weeks in summer 2008), (El Haddad et al., 2011a, 2011b, 2013). The seasonal variations and sources of aerosol have been well documented through the offline analysis of daily filter samples collected over 1 year (Bozzetti et al., 2017; Salameh et al., 2015, 2018) but this methodology only gives poor temporal resolution compared to online instruments and therefore cannot capture the fast changes in concentration and chemical composition.

In this context a new atmospheric urban background supersite dedicated to the long-term and real time chemical and physical characterization of submicron aerosol was recently implemented in Marseille. This supersite gathers state of art instruments for the measurement of aerosols (chemical composition and size distribution) and a suite of instruments for the monitoring of regulated pollutants ($PM_{2.5}$, $PM_{10}$, $NO_x$, $O_3$ and $SO_2$). The goal of the present paper is to characterize the long-term phenomenology of submicron aerosol in a coastal city. The seasonal variations, diurnal profiles, and geographical origins of $PM_1$ are presented with a focus on local and long-range pollution episodes when PM exceedance days occur. Also, the chemical characteristics of shipping/industrial emissions in summer are explored through backtrajectories analysis.

## 2 Instrumentation and Methodology

### 2.1 Marseille Supersite

Marseille-Longchamp supersite (MRS-LCP) is in the downtown park “Longchamp” (43°18′18.84″N; 5°23′40.89″E; 71 m a.s.l.) of Marseille. This site, run by a joint effort between the LCE and the French regional air quality network (AtmoSud, https://www.atmosud.org), gathers a complete set of instruments for the measurement of both regulated and non-regulated pollutants (see section 2.2). This infrastructure is intended to become a high-level research platform for monitoring air pollution and contributing to collaborative research programs. It is classified as urban background site based on the criteria used by the European Environment Agency (Larssen et al., 1999). Marseille is the second most populated city in France with more than 1.8 million of inhabitants. The distance driven by vehicles was 2.4 billion km within a 5 km radius around the supersite in 2017, which was almost 10 times higher than the distance travelled per km over the total French road network. The city hosts

the second largest harbour of the Mediterranean Sea and the first French harbour, with almost 4000 ship berthing in the several basins of Marseille for the year 2017. At 40 km North-West of the city is located the large industrial complex of Fos-sur-mer with petroleum refining, shipbuilding, steel facilities, and coke production plants (El Haddad et al., 2011b; Salameh et al., 2018). The region is well-known for active photochemistry inducing high ozone concentrations (Flaounas et al., 2009) during summer periods (Figure 1), and frequent secondary organic aerosol formation events (El Haddad et al., 2013). Air mass circulation is complex in Marseille area (Drobinski et al., 2007; El Haddad et al., 2013; Flaounas et al., 2009) and is held by the surrounding topography. The city is bordered by Mediterranean Sea from the southwest and enclosed from the north, east and south by mountain ranges up to 700 m a.s.l. The synoptic air masses come from the Rhone valley, the Atlantic and Mediterranean Sea (Drobinski et al., 2007). Moreover, Marseille air quality is often affected by the Mistral wind and sea/land breeze cycles. The Mistral is a strong wind blowing from the North-West (310°-360°) along the lower Rhône River valley toward the Mediterranean Sea. The South-Westerly sea breeze (190°-270°) and North-Easterly land breeze (5°-90°) are local winds prevailing during weak Mistral wind (Figure S1). Land breeze circulation is often associated with high pollution levels over Marseille due to the low pollutants dispersion. In the early morning of summer days, Marseille is directly downwind of the industrial area and the harbour basins. As the temperature of the land surface rises, sea breeze sets in and the polluted air masses from the industrial area are transported over the Mediterranean Sea before reaching the city.

Figure 1 shows annual average concentrations for several pollutants ($NO_x$, $O_3$, $SO_2$, $PM_{10}$, $PM_{2.5}$, BC) measured over the last 11 years at MRS-LCP station. The right graphs display the sorted daily concentrations and exceedance days according to WHO recommendations for the years 2017 and 2018. For most of these pollutants the annual concentrations have slowly decreased through the time except for $O_3$ in a similar way as observed in Europe (European Environment Agency, 2019). Around 144 days above the 8h-average threshold of 100 µg $m^{-3}$ (WHO) were recorded in 2017 and 2018, mostly during summertime. Averaged $SO_2$ concentrations display a decreasing trend thanks to EU legislation on emission control and lower fuel sulphur content (limited to 50 ppm in 2005 and then to 10 ppm since 2009). Another explanation could be the decline of maritime transport during several years since the 2009 financial crisis. But from 2013, however, $SO_2$ concentrations seem to increase again and could be linked to the enhancement of maritime activity at Marseille harbour. Indeed the French goods maritime transport rose again between 2012 and 2017 (+5.9%), together with the passenger maritime transport that has continuously been enhanced for the past 10 years (+31.7%) (French Commissioner-General for Sustainable Development, 2019). For particulate matter ($PM_{10}$, $PM_{2.5}$), annual concentrations show very slight decreases in the last 11 years but remain above the WHO recommendations. This is consistent with the global decrease in the EU-28 since 2000 (27 and 28% reduction for $PM_{10}$ and $PM_{2.5}$, respectively) and the $PM_{2.5}$ target value was exceeded in 21 countries (European Environment Agency, 2019). In the last 2 years about 12 and 28 days with exceedance concentrations were recorded respectively for $PM_{10}$ and $PM_{2.5}$, mainly in winter periods.

### 2.2 Instrumentation and data analysis

#### 2.2.1 ACSM sampling and data corrections

Ambient submicron particles (NR-$PM_1$) were measured continuously from 1 February 2017 to 13 April 2018 using a time-of-flight aerosol chemical speciation monitor (ToF-ACSM, Aerodyne Research Inc., USA). The instrument provides quantitative assessment of non-refractory species as organics, nitrate, sulfate, ammonium and chloride in the size range 40-1000 nm. The aerosol is sampled at the main inlet at a flow rate of 3 L $min^{-1}$ and dried using a Nafion dryer system (Perma Pure, New Jersey, USA) to keep the relative humidity (RH) below 40%. A subsample flow of 0.085 lpm passes through a critical orifice and enters an aerodynamic lens that focuses the particles into a narrow beam, these are then flash-vaporized upon impaction on a heated tungsten plate at 600°C. The resulting vapours are ionized using 70 eV electron impact (EI) ionization. The time-of-flight mass spectrometer (ETOF, TOFWERK, Thun, Switzerland) provides mass spectra at a mass-to-charge resolution of M/ΔM=600. The data were acquired at a time resolution of 15 min using Igro-DAQ v.2.1.4 software and by Tofware v.2.5.13 written in Igor Pro (Wave Metric inc., Lake Oswego, Oregon, USA). Further description and detail of the instrument are presented by Fröhlich et al. (2013, 2015) and Timonen et al. (2016). Calibrations of ionization efficiency (IE) of nitrate and relative ionization efficiency (RIE) of ammonium and sulfate were repeated 3 times over the 14-month measurement period. The calculated values are summarized in table S1. Table S2 lists the detection limits, calculated as three time the noise level, for the 5 quantified species. The collection efficiencies values (CE) were corrected using algorithms described by Middlebrook et al. (2012), the time-dependant CE are shown in Figure S2. For this dataset CE is assessed as 0.47±0.05 which is comparable to values typically found for ambient aerosol (0.5, Middlebrook et al., 2012). An overall uncertainty of ± 30% is associated to the total mass concentrations. It includes the uncertainties on the IE, RIE and CE values (Bahreini et al., 2009).

The organic aerosol (OA) mass was corrected to account for measurement interferences. According to Pieber et al. (2016) ammonium nitrate induces an overestimation of OA at m/z 44. A correction is introduced in the fragmentation table by measuring the relationship between measured $CO_2^+$ and the $NH_4NO_3$ mass measured during ToF-ACSM calibrations (see equation S1). Our dataset showed very little contribution of $NH_4NO_3$ on the organic m/z 44 with value ranging from 0.1-0.5% (table S3).

The ion fragments at m/z 30 and m/z 46 assigned to nitrate ($NO^+$ and $NO_2^+$) may contain interferences from organic species like $CH_2O^+$ at m/z 30 and $CH_2O_2^+$ at m/z 46. These interferences lead to an overestimation of UMR nitrate and can falsely suggest the presence of organic nitrate in high OA/$NO_3^-$ environments. Here the m/z 30 and m/z 46 signals have been corrected for these interferences by using correlated organic signals respectively at m/z 29 from $CHO^+$ and m/z 45 from $CHO_2^+$ (equation S2), as recommended by Fry et al. (2018). These peaks were the closest organic signals to the nitrate peaks with organic interferences.
### 2.2.2 Collocated instruments

ACMS measurements were combined with several on-line collocated instruments. A dual spot 7-wavelenght AE33 Aethalometer (Magee Scientific) (Drinovec et al., 2015) equipped with a $PM_{2.5}$ cut-off inlet was used to measure equivalent black carbon (BC) concentrations at a 1 min time resolution. Equivalent black carbon concentrations were calculated from the absorption coefficient at 880 nm with the default mass absorption cross section (MAC) implemented in the AE33 (7.77 $m^2\ g^{-1}$). The submicron aerosol number size distribution was investigated with the model 3031 ultrafine particle monitor (TSI Inc., Minnesota, USA) for the whole study period. This instrument provides measurements from 20 to 1000 nm, with six channels of size resolution. The aerosol number size distribution in the range 10.25-1084 nm was further explored over 45 channels using a Scanning Mobility Particle Sizer system (SMPS, L-DMA, CPC5403, GRIMM) for two periods: from 23 June to 12 August 2017 (summer period) and from 6 November 2017 to 11 January 2018 (winter period).

Off-line measurements were carried out to collect particles (24h $PM_1$) onto 150 mm-diameter quartz fiber filters (Pall Gellman, TISSUQUARTZ) (8h00 to 8h00 UT) using a high volume sampler (Digitel DA-80) operating at a flowrate of 30 $m^3.h^{-1}$ . 45 filters were discontinuously sampled from 01 March to 01 May 2017 (22 filters) and from 01 July to 23 September 2017 (23 filters). These filters were analysed in order to determine the major anions and cations using ion chromatography (Sciare et al., 2008), and elemental/organic content using Sunset OC/EC analyser (EUSAAR2 thermal protocol) according to Cavalli et al. (2010).

Continuous measurements of $SO_2$, $NO_x$ and $O_3$, and $PM_x$ were carried out by the air quality monitoring station. A M100E UV fluorescence analyser, a M200E chemiluminescence analyser (Teledyne API, California, USA) and a Serinus 10 ozone analyser (Ecotech, Australia) were deployed for the $SO_2$, $NO_x$ and $O_3$ measurements, respectively. A Continuous Beta-attenuation continuous particulate monitor (BAM 1020, Met One Instruments Inc., Oregon, USA) was used to measure the mass concentrations of $PM_{2.5}$ and $PM_{10}$ and an optical particle counter (FIDAS 200, PALAS, Germany) was dedicated to the measurement of $PM_1$, $PM_{2.5}$ and $PM_{10}$ since February 2018. All the time resolutions of this instrumental panel were synchronized to 15 min.

### 2.3 Meteorological data and backtrajectories analysis

The site is equipped with an anemometer sonic 3D to provide temperature and wind measurements (directions and speeds) according to 3 orthogonal axes. Precipitations and relative humidity parameters were taken from the Vaudrans meteorological station located 6 km away for the MRS-LC site (43°18′26″N; 5°28′28″E). Non-parametric wind regression (NWR; Henry et al., 2009) and sustained wind incidence method-2 (SWIM-2; Olson et al., 2012) algorithms were used to generate pollution roses. NWR and SWIM-2 analyses were performed using the ZeFir toolkit (Petit et al., 2017a). To investigate air mass origin during specific pollution episodes, 72h-backtrajectories were calculated every hour from the PC-based version of HYSPLIT (Draxler et al., 1999) with weekly 1° Global Data Assimilation System (GDAS) meteorological field data. Backtrajectories were set to end at MRS-LCP coordinates (43°18′18.84″N;5°23′40.89″E; 64 m a.g.l.). Planetary Boundary Layer (PBL) height

data were directly extracted from the GDAS files with HYSPLIT. Cluster analyses were then applied on the calculated backtrajectories for the summer 2017 period. Finally, Concentration Weighted Trajectory (CWT) was performed with ZeFir to investigate the locations most frequently associated with elevated concentrations. CWT approach couples pollutant concentrations measured at a receptor site (MRS-LCP) with backtrajectories and displays the localisation of air masses associated with high concentrations at the site (Ashbaugh et al., 1985; Petit et al., 2017b). For this study, the CWT domain was set in the range of (40-46° N; -5-10° E) with the grid cell size of 0.05°×0.05°.

## 3 Results and discussion

### 3.1 Cross-validation of $PM_1$ chemical species concentrations

The $PM_1$ mass concentrations measured by the ToF-ACSM were compared with 24h-$PM_1$ filter measurements (Figure S3). The ACSM concentrations were daily averaged and compared with respective offline measurements from 1 March to 23 September 2017 (n=46). A good agreement is found for ammonium and sulfate ($R^2$=0.71 and slope of 0.84, and $R^2$=0.76 and slope of 0.89, respectively). For nitrate the results are less consistent ($R^2$=0.69 and slope of 1.22). This higher slope can be attributed to the volatilization of nitrate from the filters during hot periods (Ripoll et al., 2015; Schaap et al., 2004a). The $NH_{4\ measured}/NH_{4\ predicted}$ ratio was also investigated from 1 February 2017 to 13 April 2018 (Figure S4) as an indirect proxy for particle acidity (Zhang et al., 2007a) and/or presence of high organic nitrate concentrations (Petit et al., 2017b). The $NH_{4\ predicted}$ represents the theoretical ammonium concentration needed to neutralize the inorganic species concentrations ($NO_3^-$, $SO_4^{2-}$, $Cl^-$). For most of the cities inorganic species tend to be fully neutralized and a limited amount of acidic particles can only be observed when the site is located close to an emission source (e.g. industries, harbour, fire event…). Here the slope value close to 1 (0.95) reflects a full neutralization for all anions and suggests that there is enough ammonia in the gas phase to neutralize these species despite the nearby harbour and industrial complex.

The OC to organics comparison (filters vs ACSM concentrations) showed a good correlation with $R^2$=0.79 with a slope (corresponding to the OM-to-OC ratio) of 1.9. This value is slightly higher than the recommended values for urban areas (1.6±0.2, Petit et al., 2015; Stavroulas et al., 2019; Turpin and Lim, 2001). It is possible that the chosen sampling periods for the comparison (spring and summer 2017) bias high the OM-to-OC value as the photochemical activity and thus atmospheric aging are expected to increase. Ratio up to 2.2 have been observed when a significant fraction of particulate matter is made of aged aerosol (Aiken et al., 2008; Minguillón et al., 2011).

BC measurements from AE33 are compared with EC offline filters and an excellent agreement is found with $R^2$=0.87. The slope of 1.52 relates the difference in measurements properties between EC (thermic) and BC (absorption). The difference between the measured BC and EC could be attributed to the variability of the mass absorption coefficient (MAC) value used to convert the absorbance to BC mass concentrations in the AE33 instrument. This value could be influenced by light-absorbing OC like brown carbon from biomass burning. Here the slope is in agreement with values from other studies: 1.14 to 2.13

through one year at Fresno supersite (Park et al., 2006); 1.62 and 1.92 using EUSAAR2 and NIOSH870 protocols, respectively, between January 2015 and July 2016 at the Environment-Climate Observatory of Lecce (Merico et al., 2019).

Finally the sum of ACSM species and BC mass concentrations were compared to the estimated mass using SMPS volume during the two deployment periods of this instrument (from 23 June to 12 August 2017 and from 6 November 2017 to 11 January 2018). For the SMPS mass conversion, a density ($d_{calc}$) was estimated taking into account the chemical composition of $PM_1$ with respective densities of 1.2 g cm$^{-3}$ for organic matter, 1.75 g cm$^{-3}$ for nitrate, sulfate and ammonium, 1.52 g cm$^{-3}$ for chloride (Cross et al., 2007). The organic aerosol density can increase whether there are high contributions of carboxylic/dicarboxylic acids (1.46±0.23 g cm$^{-3}$) and/or polycyclic aromatic hydrocarbons (1.28±0.12 g cm$^{-3}$). In contrast this density would decrease with high contributions of n-alkanes (0.79±0.01 g cm$^{-3}$) and/or n-alkanoic acids (0.89±0.07 g cm$^{-3}$) (Turpin and Lim, 2001). A default density of 1.77 g cm$^{-3}$ is applied for BC as recommended by Poulain et al. (2014). The calculated density is based on the following equation (Salcedo et al., 2006):

$$d_{calc} = \frac{[Total_{ACSM} + BC]}{\frac{[NO_3^-]+[SO_4^{2-}]+[NH_4^+]}{1.75} + \frac{[Cl^-]}{1.52} + \frac{[OM]}{1.2} + \frac{[BC]}{1.77}}, \quad (1)$$

where the time-dependant mass fractions are based on ACSM and AE33 measurements. The density was found to range between 1.24 and 1.77, with an average value of 1.43. The scatter plot of ACSM+BC concentrations vs. $PM_1$ concentrations from SMPS in Figure 2a shows strong correlation ($R^2$=0.81) and slope close to unity (slope=1.02 and intercept=0.46 for an orthogonal regression). Reconstituted mass (ACSM+AE33) was also compared to $PM_1$ mass measurements from FIDAS for the last 3 months of the database (19 February to 13 April 2018) in Figure S3. Again satisfactory results are displayed with $R^2$=0.89 and slope=0.96 (here the intercept is automatically set to 0) and show the consistency of the different measurements. The linear regression analysis of reconstituted $PM_1$ vs $PM_{2.5}$ yielded a slope of 0.88-0.89 for a confident interval of 99% and $R^2$=0.77 (Figure 2b) so we can assume that $PM_{2.5}$ concentrations are mainly composed of submicron particles.

### 3.2 Long-term trend of submicron aerosol

#### 3.2.1 Seasonal patterns

Time series and seasonal contributions of ACSM components, Black Carbon and daily-$PM_{2.5}$ are shown in Figure 3. A summary of the seasonal statistics (average and standard deviation) is reported in table 1. The averaged concentration was 9.9 µg m$^{-3}$ and 12.0 µg m$^{-3}$ for $PM_1$ and $PM_{2.5}$, respectively. The mass concentrations measured by the BAM 1020 show a mass exceedance for this period at Marseille according to WHO recommendation (10 µg m$^{-3}$). The highest concentrations for $PM_1$ were measured during winter with 11.9 µg m$^{-3}$ and the lowest in spring with 8.1 µg m$^{-3}$. Several peaks reached 50 µg m$^{-3}$ especially during cold periods in winter and autumn. The averaged $PM_1$ chemical composition over the period is dominated by organics, (49.7%), and BC (17.1%) while sulfate accounts for 14.6%, nitrate for 10.2%, ammonium for 7.9% and chloride for 0.5% only. Past long-term studies on daily $PM_{2.5}$ filters in MRS-LCP described similar yearly trend for those species

(Bozzetti et al., 2017; Salameh et al., 2015): 44-46% for OM, 13% for BC (as EC measurements were performed during these studies, the value was multiplied by the BC/EC slope found in the present work to trace back to BC), 11% for sulfate, 8% for nitrate, 7% for ammonium and less than 1% for chloride over the August 2011 to July 2012 period. This submicron aerosol composition is comparable to other Mediterranean coastal cities, where a noticeable proportion of sulfate is generally observed (Minguillón et al., 2015; Salameh et al., 2015; Stavroulas et al., 2019), whereas most urban sites in northern/central Europe show higher nitrate contribution (Lianou et al., 2011; Petit et al., 2015; Young et al., 2015). This is especially true in summer, where the highest contribution of sulfate to the total mass of $PM_1$ is observed (24.1%). The present study confirms what had been previously reported during a short time summer measurement campaign with a C-ToF-AMS (El Haddad et al., 2013). El Haddad et al. (2013) noticed that elevated sulfate periods corresponded to air masses transported from the Mediterranean Sea. The origin of sulfate aerosol is further examined in section 3.3.2.

The average carbonaceous fraction (OA+BC) contributes to 66.8% of the $PM_1$ mass. By converting BC to EC with the previous slope of 1.52 (see section 3.1.) the carbonaceous fraction would be 64.8% which is high compared to the OM+EC contributions of $PM_{2.5}$ from several urban sites in Europe (26 to 47%; Putaud et al., 2010). The OA dominance in every season is a common feature observed for European urban areas (Zhang et al., 2007b). Highest averaged values of 6.2 µg $m^{-3}$ and 5.1 µg $m^{-3}$ are observed during winter and autumn months respectively (Figure S5). This trend was also observed in the 2011-2012 period but higher concentrations were found, with 12.1 µg $m^{-3}$ in winter and 11.1 µg $m^{-3}$ in autumn (Bozzetti et al., 2017). The year 2011 showed particularly large amount of days with high level of $PM_{2.5}$ in the autumn-winter period (43 days exceeding WHO $PM_{2.5}$ threshold value; table 2) which may explain this difference. BC seasonal cycle has a similar pattern with highest values of 2.1 µg $m^{-3}$ in winter and 1.9 µg $m^{-3}$ in autumn. These high carbonaceous concentrations during the cold months are expected considering i) a reduction of the planetary boundary layer (PBL) height resulting in the accumulation of pollutants at ground level compared to other seasons and ii) increasing emissions from residential heating and open combustion of green wastes (Bozzetti et al., 2017). The deconvolution of BC into two contributions, fossil fuel and wood burning (respectively $BC_{FF}$ and $BC_{WB}$) was carried out using the aethalometer model (Sandradewi et al., 2008). As a first step the procedure recommended by Zotter et al. (2017) (i.e. to use the 470 and 950 nm wavelengths with an Angström exponent of 1.68 and 0.9 for pure wood burning and fossil fuel respectively) was applied. Using the suggested values led to unrealistic high $BC_{WB}$ contributions in the summer (18%) when biomass burning is expected to be negligible during the hot period. It is hypothesised, as previously suggested by Titos et al. (2017), that a fraction of $BC_{FF}$ was wrongly attributed to wood burning as a consequence of a failure of the model to reconstruct sources when the biomass burning fraction is very low. This potential bias was investigated on fossil fuel-derived $PM_1$ from a urban traffic site (station "Kaddouz", location: 43°34'49.8" N;5°37'49.3" E) during summer time. This kerbside site is located at the portal of a tunnel in the surrounding area of Marseille. In order to inspect the different combinations of Angström exponent for fossil fuel and wood burning ($\alpha_{FF}$ and $\alpha_{WB}$, respectively) a sensitivity test was performed by scanning combination changes in a $\alpha_{FF}$ range of 0.9-1.1 and a $\alpha_{WB}$ range 1.6-2 with a step size of 0.01.The set of combinations was evaluated and optimized based on the $BC_{WB}$ diurnal cycles, which were categorized according to a k-means

clustering analysis. Results are further described in the Supplement. From this analysis, a selected $\alpha_{FF}$ of 1.02 would be more representative of fresh traffic emission in Marseille and Garg et al. (2016) suggested that $\alpha_{FF} > 1$ could be more appropriate for older vehicles operating with poorly optimized engines. For $\alpha_{WB}$ , the reference value of 1.68 from Zotter et al. (2017) has been used for this study.

Seasonal results of BC contributions in MRS-LCP are shown in Figure 4 and statistic averages and standard deviations are reported in table 1. $BC_{WB}$ shows a clear seasonal variability like OA with the highest contribution to the total BC in winter (28%). Despite the mild climate in South of France, wood heating is an important source of pollution which can be explained by the poor insulation in houses. Almost 15% of housing use wood for heating in Aix-Marseille Metropolis (use of chimneys, wood stove or wood boiler) (ADEME, 2009) and around 50% of the identified equipment are still considered as non-efficient in France (ADEME, 2020). Additionally, several wood-fired heating plants located in Aubagne, Gardanne and Aix-en-Provence (15, 18 and 25 km from Marseille, respectively) might contribute to biomass burning emissions. Besides wood heating, the open burning is expected to contribute to the increasing OA+$BC_{WB}$ concentrations, particularly in autumn. This includes the agricultural burning, since one third of the department surface area is dedicated to agricultural activities, but also the green waste burning. While this latter practice is prohibited, it is suspected to still be used nowadays. Summer $BC_{WB}$ in MRS-LCP represents the lowest contribution with 7%, which is in the same range than in Fos-sur-Mer (2%; Bonvalot et al., 2019) and another big Mediterranean coastal city like Athens (6%; Diapouli et al., 2017).

The $BC_{FF}$ is still the main contributor to ambient black carbon (72-93% of total BC) with values ranging from 1.1 to 1.7 µg $m^{-3}$ throughout the seasons. The elevated BC concentrations could be explained by the proximity of the monitoring station to the city center and thus to local urban emissions. Vehicular emissions highly contribute to primary carbonaceous fraction in Marseille as demonstrated by El Haddad et al. (2011b) and can be a significant source for OA and $BC_{FF}$. Even if $BC_{FF}$ is assumed to be an excellent marker of vehicular emissions (Herich et al., 2011), in Marseille other sources as oil-fired boilers, industrial and shipping activities could contribute to total $BC_{FF}$.

Nitrate exhibits maximum contribution in winter and early spring (13.2% and 14.0%, with 1.58 and 1.13 µg $m^{-3}$ respectively) and minimum contribution during summer (2.6% with 0.24 µg $m^{-3}$).This feature has been already observed in other European cities (Minguillón et al., 2015; Petit et al., 2015; Reyes-Villegas et al., 2016; Young et al., 2015). Bozzetti et al. (2017) observed lower fraction in spring possibly because of the volatilization of nitrate from the filters surface in warm conditions. Several episodes were reported mostly in late winter/early spring (described by elevated nitrate dispersion for February and March in Figure S5) and concentrations of other species also increased during these nitrate events conducting to highly polluted days. These episodes are investigated in section 3.3.1. Nitrate was further categorised into inorganic and organic fraction using the method described by Farmer et al. (2010). The detailed methodology is presented in SI.

Seasonal mass fraction of $NO_{3,Org}$ and $NO_{3,Inorg}$ are displayed in Figure 4. $NO_{3,Org}$ average concentration ranges from 0.09 µg $m^{-3}$ in summer to 0.26 µg $m^{-3}$ in winter. The resulting average $NO_{3,Org}$ fraction for the whole dataset is 20±7%. The error here is determined from error propagation calculations described by Farmer et al. (2010) and is detailed in SI. This fraction is in the range of reported values for European urban sites (28%, Mohr et al., 2012; 24%, Saarikoski et al., 2012). The highest $NO_{3,Org}$

contribution happens to be in summer (38%) when the total nitrate concentration is at its lowest level. $NO_{3,Org}$ is produced from volatile organic compounds (VOCs) oxidations by nitrate radicals and photochemical oxidation with $NO_x$. Biogenic VOCs emissions are high in summer in Mediterranean area (Parra et al., 2004; Steinbrecher et al., 2009) and provide an important potential source for organic nitrate formation.

$NO_{3,Org}$ takes into account only the nitrate functionality of organic nitrates. Assuming a molecular weight between 200 and 300 g $mol^{-1}$ for particle organic nitrates, contribution of organic nitrates to total OA can be estimated (Xu et al., 2015). Results presented in Table S4 give a contribution to total OA of about 6–10% in summer, 11-17% in autumn, 14-20% in winter and 18–28% in spring. Highest contributions in springtime can be due to increased level of biogenic VOCs coupled with favourable meteorological conditions for partitioning. These results suggest that despite the low contribution of $NO_{3,Org}$ to total nitrate, organic nitrates can be an significant fraction of OA in Marseille.

The $NO_{3,Inorg}$ average seasonal concentrations range from 0.15 µg $m^{-3}$ in summer to 1.31 µg $m^{-3}$ in winter. Here $NO_{3,Inorg}$ was assumed to be mostly ammonium nitrate particles. Ammonium nitrate is semi volatile and its gas/particle partitioning is affected by temperature and relative humidity changes (Stelson and Seinfeld, 1982), which lead to enhanced particle partitioning during winter and higher evaporation during summer (Huffman et al., 2009). This is supported by Figure S5 where lowest temperature and highest relative humidity values are found in winter and early spring, while the inverse trend is illustrated for summer months.

Winter and springtime ammonium concentrations can be driven by the greater availability of ammonia from agricultural activity and waste management. Moreover Suarez-Bertoa et al. (2015) mentioned that urban traffic emissions of ammonia have increased in Europe (+378%) over the last decades leading to possible enhanced ammonium concentrations. High concentrations of inorganic nitrate are related to elevated levels of $NO_x$ in winter. This is described in Figure 1 where the hourly maximum concentrations of $NO_x$ emissions mostly appear in winter seasons.

It is well known that ultrafine particles (UFPs; diameter <100nm) do not affect the mass concentrations but might be relevant for health-related issues, and their long-term variability is explored in this study. The total average UFPs number concentration (20-100 nm) measured with a TSI 3031 monitor over the full study was 7765 $cm^{-3}$. These values are similar to those observed in Barcelona but are higher than those found in Prague, Madrid and Rome (Borsós et al., 2012; Brines et al., 2015). In Marseille, UFPs represented at 85% of total submicron particle number, in agreement with previous observations in urban environment (Rodríguez and Cuevas, 2007; Wehner and Wiedensohler, 2003). UFPs average concentration were slightly higher in winter and autumn with average values of 8600 $cm^{-3}$ and 8100 $cm^{-3}$, respectively, while an average value of 7500 $cm^{-3}$ was found in spring and summer. The seasonal variation followed the general patterns observed for BC and OA concentrations over the year, suggesting that most particles in number arise from sources of combustion. However, when considering size distribution measurements down to 10-nm, SMPS data revealed the occurrence of sharp UFPs events more frequently in the summer than in winter, with 10 events exceeding 50 000 particles $cm^{-3}$ in summer against only 1 in winter (Figure 3). To get insights into the sources and processes contributing to ultrafine particles in urban ambient air, number concentrations and BC were

investigated using the methodology developed by Rodríguez and Cuevas (2007). This methodology has been extensively applied in urban environment to apportion the number concentration of primary and secondary sources (del Águila et al., 2018; González et al., 2011; Hama et al., 2017a; Hama et al., 2017b; Reche et al., 2011; Rodríguez and Cuevas, 2007; Tobías et al., 2018). To refine the method, $BC_{FF}$ was used instead of total BC to better apportion primary traffic emissions. The total measured UFPs number concentration (N) can be splitted in two components:

$$N_1 = S_1 . BC_{FF} , \quad (2)$$

$$N_2 = N - N_1 , \quad (3)$$

where $N_1$ accounts for fresh primary emissions of vehicle exhaust, directly emitted in the particle phase or nucleating immediately after emission (Arnold et al., 2006; Burtscher, 2005; Kittelson, 1998) and $N_2$ accounts mostly for secondary particles formed in the atmosphere during the dilution and cooling of the exhaust emissions. $S_1$ is the slope estimated using best-fit line to the points aligned in the lower edge of N vs $BC_{FF}$ scatter plot (see Figure S7 for details).

$S_1$ was calculated for each season by fitting data below the $10^{th}$ percentile of N/$BC_{FF}$. The $N_1$ and $N_2$ fractions were derived from the TSI 3031 measurements and average number concentrations during each season are reported in table 1 and Figure 4. The $N_2$ fraction was predominant with number concentrations 1.02 to 1.7 times higher than $N_1$. The $N_1$ seasonal trend slightly varied through the measurement period with a contribution between 37 and 50%. These emissions appear to be dominant in winter and autumn, with average concentrations of ≈ 4200 and 3800 $cm^{-3}$. An explanation for these results could be either the higher traffic rate or the low temperature influencing the soot particle formation during combustion in these seasons. In contrast, the $N_2$ average particle numbers were higher in spring and summer as the concentrations reached its highest value (≈ 4300 $cm^{-3}$ and 4800 $cm^{-3}$). This behaviour will be further addressed in the following section.

### 3.2.2 Diurnal profiles

Figure 5 shows the average diurnal profiles of OA, $NH_4^+$, $NO_3^-$, $SO_4^{2-}$, $BC_{FF}$ and $BC_{WB}$ across the different seasons. Distinctive diurnal patterns are found for carbonaceous aerosols. First, a clear traffic-related diurnal profile is observed for $BC_{FF}$ with a morning peak and an evening peak starting at the typical rush hours (04:00 UTC and 16:00 UTC respectively). The amplitude of the traffic-related diurnal cycle seemed to be affected by meteorological conditions since it varied within the year, with maximum min to max amplitude of 0.9 during winter and minimum min to max amplitude of 0.5 during summer. The $BC_{WB}$ had a significantly different diurnal cycle with a typical increase starting at 17:00 UTC and a maximum level of 1.1 µg $m^{-3}$ at night-time.

The mean to max amplitude increased steadily from spring, autumn to winter (0.1, 0.2 and 0.4, respectively), following the increased heating demand. Under specific meteorological conditions (no rain, low wind speed, low boundary layer) this source led to the highest levels of $PM_1$ episodes in Marseille. This is discussed in detail in Sect. 3.3.1. In winter and autumn, OA diurnal cycle mainly resulted from the superposition of both traffic-related and wood burning-related cycles with maxima of

4.9 µg m$^{-3}$ and 7.9 µg m$^{-3}$ for the morning and evening peak, respectively. In spring and summer an additional local maximum appeared during mid-day (3.8 µg m$^{-3}$). While this peak may partly be related to a distinct local source like cooking emissions (Bozzetti et al., 2017), formation of secondary organic aerosol is also expected, as observed by El Haddad et al. (2013).

The diurnal variations of $NO_{3,Org}$ are represented for each season in Figure 5. Concentrations increased after sunset and were higher at night, likely due to the oxidation of VOCs by the nitrate radical (Kiendler-Scharr et al., 2016). This trend was more pronounced in summertime with enhanced biogenic VOCs emissions. Summer $NO_{3,Org}$ profile was investigated for June 2017 when concentrations were above the detection limits. Slight daily enhancements of $NO_{3,Org}$ were found in the morning (around 08:00 UTC) and are attributed to photooxidation of VOCs in the presence of high nitrogen monoxide NO concentrations (the diurnal profiles are also reported in Figure 5 and show maxima at 07:00 UTC) as mentioned by Xu et al. (2015). Similarly, $NO_{3,Inorg}$ diurnal pattern suggests fast formation mechanism from local $NO_x$ emissions. Concentrations are higher during night-time when the condensation to the particle phase is favoured by meteorological conditions (low temperature and high relative humidity).

Sulfate and ammonium exhibit very similar profiles, except for winter and early spring when ammonium is mainly associated to nitrate. The quite flat diurnal profiles of ammonium sulfate during autumn, winter, and spring might be due to the regional character of this component (regional transport) (Seinfeld and Pandis, 2016).

In summer however, ammonium sulfate diurnal profile shows a clear increase during the daylight period with maxima reached at noon, possibly due to local photochemical production of sulfate from its precursor $SO_2$ emitted by nearby shipping and industrial activities and advected to the city as the sea breeze sets in. Some studies suggest fast $SO_2$ to sulfate conversion in the exhaust plumes (Healy et al., 2009; Lack et al., 2009). This local sulfate fraction will be discussed in detail in Section 3.3.2.

Average daily profiles for $N_1$ and $N_2$ are represented in Figure 5 and show some discrepancies. Particle number $N_1$ exhibits maxima during morning and evening traffic rush hours similarly to NO concentrations, when UFPs are mainly associated to vehicle exhaust emissions. The $N_2$ daily profiles display different diurnal patterns: in winter $N_2$ follows the traffic bimodal profile with 1-2 hours shift (morning peak at starting at 06:00 UTC and evening at 17:00 UTC) which can highlight the fast homogeneous dilution/cooling (favoured by the low temperature and high relative humidity conditions) and mixing of the vehicle exhaust in the ambient air (Casati et al., 2007; Charron and Harrison, 2003). In summer, $N_2$ exhibits a daily evolution with a broad maximum during daylight. This trend closely follows the daily evolution of temperature, ozone and $SO_4^{2-}$ concentration suggesting a photo-oxidative process of gaseous precursors (Woo et al., 2001), as $SO_2$, combined with more dilution of pollutants when the boundary layer increases (Reche et al., 2011). For the transitional seasons (autumn and spring) the patterns reveal some mixing between the cars exhaust cooling and photochemistry states and $N_2$ can be attributed to the two processes.

In order to investigate the UFPs apportionment below 20 nm the methodology was applied to SMPS data during the available summer period for a range between 10 and 20 nm. The $N_{2\ (10\text{-}20\ nm)}$ number concentration, corresponding to 90% of the total number in this range, showed the same trend of $SO_2$ diurnal evolution (Figure 5). The $SO_2$ is considered a gas tracer for

industrial and shipping activity and it was advected to the monitoring station from the morning. Furthermore, both $SO_2$ and $N_{2\,(10\text{-}20\,nm)}$ concentrations increased suggesting the nucleation of sulfuric acid particles (Burtscher, 2005; González et al., 2011). Overall the UFPs investigation demonstrates that secondary particle formation is an important contributor to particle number in Marseille and besides road traffic, there is some high influence of industrial/shipping mixed sources.

### 3.3 Case Study of $PM_1$ polluted periods

#### 3.3.1 PM episodes with exceedance days: local vs long-range transport

Table 2 summarizes the number of $PM_{2.5}$ exceedance days occurring during the last decade in MRS-LCP, using as reference value 25 µg $m^{-3}$ 24-hour mean from the WHO guideline for particulate matter. Overall, the number of exceedance days strongly declined over the years (it was 3-4 times lower in 2017 than in the 2009-2011 period) but the seasonal trend remained similar with most frequent exceedance days in winter and early spring. $PM_{2.5}$ polluted episodes were scarce in summer except for 2 outstanding years in 2009 and 2010 where 13 and 19 exceedance days occurred. In the present study, 15 exceedance days were monitored over the 14 month-period (2017/02/01 – 2018/04/13). These exceedance periods occurred between November 2017 and March 2018, in good agreement with what observed in the last decade, and they could last from 1 to 4 consecutive days (Figure 1). It was shown before that $PM_1$ represent around 88% of $PM_{2.5}$ in Marseille and exploring this fraction is assumed to be representative of $PM_{2.5}$ polluted events. Because of similarities between different events, the exceedance days could be gathered into two categories, corresponding to local event and long-range event. Then two pollution episodes have been selected for a deeper investigation: a local Christmas event (23-24 December 2017) and a long-range event (22-25 February 2018).

Figure 6 shows the time series of $PM_1$ chemical composition and the meteorological parameters (i.e. temperature, relative humidity, wind direction, PBL height and precipitations) during these 2 events. An assessment of transport dynamics of aerosols was carried out using the $BC/SO_4^{2-}$ ratio as suggested by Petit et al. (2015). The sulfate fraction ($SO_4^{2-}$) is considered a good tracer for long-range transport, whereas BC refers to local influence from the city and the surrounding area. The use of this local/long-range proxy requires the assumption of minor local source of $SO_2$ and no direct $SO_4^{2-}$ formation in plumes at a local scale during the 2 episodes. Therefore the industrial and shipping contributions to global pollution have to be low for these periods**.** This is true except for a short period on the 23rd of December around 12:00 UTC when a temporary increase of $SO_4^{2-}$decreases the $BC/SO_4^{2-}$ ratio. This short event possibly originated from industrial and/or shipping emissions as $SO_2$ and UFPs concentrations also increased. Additionally, aged air masses might contain BC particles (Laborde et al., 2013) and this interference must be considered during the $BC/SO_4^{2-}$ ratio analysis.

The Christmas local episode presented two consecutive exceedance days for $PM_{2.5}$ concentrations: the 23rd and 24th December 2017 (Figure 6a). $PM_1$ (ACSM+BC) average concentration was 28.8 µg $m^{-3}$ and was dominated by OA (63%) and BC (21%). Carbonaceous concentrations increased mostly in the evening by a factor of 4 to 5 in the space of a few minutes. The temporal

evolution of BC contributions (Figure 6a) clearly indicated a predominance of $BC_{WB}$ during these nights with a contribution reaching sometimes 100%. At night PM size distribution ranged between 70 and 200 nm, typical of wood burning emissions (Coudray et al., 2009). High $BC/SO_4^{2-}$ ratio (average of 6.12 with values up to 20) and decreasing PBL height (470 m) suggested a strong local influence**.** Figure 7 displays NWR analysis plot for OA and $BC_{WB}$ and show evidence of high concentrations under North-East land breeze. This wind analysis is also performed on the fraction of $C_2H_4O_2^+$ organic ion (f60) which is a pure tracer of levoglucosan fragmentation (Alfarra et al., 2007) and thus of biomass burning emissions. The results displayed a similar hotspot than OA and $BC_{WB}$ concentrations.

This configuration is very frequent in Marseille, as land breeze prevailed for around 25% of time, as shown in Figure S8. Under such meteorological conditions, the average annual $PM_1$ concentration was 14.3 µg $m^{-3}$ instead of 8.49 µg $m^{-3}$ for the remaining period. These winds transport to the city anthropogenic emissions from the surrounding suburban residential areas of Marseille, in the north-easterly direction. In winter, these areas are a few degrees colder than the city of Marseille, resulting in an increased use of firewood as an auxiliary heating source. After the 25th December, the $PM_1$ levels dropped down as the wind direction shifted from North-East to East/South-East. The $PM_1$ decrease was also favoured by the vertical dilution that occurred with increased PBL height combined with precipitations.

The February long-range event took place during a period of four consecutive $PM_{2.5}$ exceedance days from 22$^{nd}$ to 25$^{th}$ February 2017 (Figure 6b). The average $PM_1$ concentration was 31.2 µg $m^{-3}$ and the aerosol chemical composition was stable with OA contribution of 41%, followed by $NO_3^-$ (25%), BC (12%), $NH_4^+$ (11%), and $SO_4^{2-}$ (10%). The mass concentrations were linked to particles distributed between 200 and 1000 nm, which is in line with size distribution in the accumulation mode of ammonium nitrate, ammonium sulfate and oxidized organic species (Canagaratna et al., 2007). This event is characterised by a low $BC/SO_4^{2-}$ ratio (average of 1.13) suggesting advection of secondary pollution, dominated by OA and ammonium nitrate. The $BC/SO_4^{2-}$ ratio can reach values of 4-5 when regional background concentrations are associated with occasional increase of local BC emissions, contributing to enhanced particle level. The geographical origin of some species was then inspected. As sustained winds come from the same direction for most of the event, wind data spikes with high standard deviation must be down-weighted to conduct a reasonable wind analysis. Instead of NWR, SWIM-2 analyses are used as they allow to use a weighting term. Figure 7 shows SWIM-2 analysis plot for $NO_{3,Inorg}$ (assuming here as ammonium nitrate). The main dominant wind sector seemed to be from North to South-West direction for this species with highest concentration from North/North-West at high wind speed and South/South-West at low wind speed. This result likely shows a combined local pollution with medium/long-range transport of secondary species during this time. In order to avoid local influences and highlight the long-range emissions, wind regression analysis are conducted on $NO_{3,Inorg}$ normalized by BC. The resulting plot clearly shows higher values for the North/North-West sector attributed to strong Mistral blow (wind speed >1.7 m $s^{-1}$) conditions. SWIM-2 analysis was also conducted on the $CO_2^+$ fraction (f44) which is, at high level, specific of oxygenated organic aerosol (OOA). Again the mistral blow conditions are well distinguished suggesting a high fraction of secondary organic aerosol during the event.
North/North-Westerly winds bring into the city polluted air masses (Figure 6b and Figure 7) from central Europe that might pass also over the Pô Valley (North of Italy) as can be observed from the 72h-backtrajectories (n=32; displayed every 3 hours) displayed in Figure S9. The Pô Valley is well known for its high levels of inorganic (Schaap et al., 2004b; Squizzato et al., 2013; Diémoz et al., 2019) and secondary organic aerosol (OOA) (Saarikoski et al., 2012). The plain is enclosed by the Alpine chain and the Apennines limiting the dispersion of pollutants and thus leading to frequent pollution events. During the winter/spring period low temperature and high humidity may favour ammonium nitrate particles formation (Schaap et al., 2004b). Moreover, intense agricultural spraying occurring in early spring might enhance NPF. Once the air masses cross the Alps they are then channelled along the Rhone Valley corridor toward the Mediterranean Sea. The low altitude of the backtrajectories (<500m), when passing by the Rhone Valley, led to the accumulation of pollutants along the air masses trajectories. A strong East/South-Est wind combined with higher temperature and higher PBL height led to the dilution of atmospheric pollutants and the return to normal conditions.

The backtrajectories of a similar event occurring from the 14$^{th}$ to the 17$^{th}$ of March are represented in Figure S9. The chemical composition was similar to that reported in the previous event (OA = 44%, $NO_3^-$ = 26%, $NH_4^+$ = 11%, BC = 9%, $SO_4^{2-}$ = 9%). Air masses were transported from the North direction through North-Estern part of France, Switzerland and North-Western part of Italy. Polluted air masses crossed continental regions known to be hotspots of ammonia emissions (the French Champagne-Ardennes region, the Swiss plateau, the Pô valley, as documented by Viatte et al., 2019), explaining partly the enhanced ammonium nitrate contribution during the winter/spring long-range events.

In term of frequency of occurrence, 40% of exceedance days account for local origin (6 days) and 60% for long-range transport influence (9 days). Globally, the combined observations of phenomenology, chemical composition and meteorology allow to accurately analyse events with exceedance mass concentrations for fine particles at MRS-LCP supersite, highlighting variable situations in an urban area affected by different air mass origins.

### 3.3.2 Sulfate origin and Shipping/industrial plumes in summer

The summertime aerosol contributes little to the exceedance of $PM_{2.5}$ air quality thresholds (Figure 1). This season accounts for only 10% of the exceedance days since 2008 and none of these occurred in 2017-2018. Still, the high solar radiation and temperature combined to the dry condition are expected to enhance the formation of secondary pollutants and their accumulation. High ozone levels often exceeding the WHO threshold are indeed recorded in Marseille (Figure 1). These conditions also affect the aerosol composition as it results in an enhanced secondary sulfate fraction (24% of $PM_1$ on average and a maximum contribution of 66%). As shown by the NWR plot in Figure 7, high concentrations of sulfate are associated to a rather broad range of wind sectors and speeds, as expected for potentially aged and processes aerosol. While the geographic area where high sulfate concentrations are observed extend between the South-West and the North-East sectors, there is still a predominance from the South-West sector. In this sector, prevalence of high concentrations of $SO_2$, the major precursor of sulfate aerosol, is also clearly observed. These $SO_2$ concentrations are associated with high UFPs plumes. Figure 8 shows the

averaged UFPs number concentrations and the seasonal frequency occurrence of peaks concentration according to $SO_2$ concentrations classes. UFPs number increases with the enhanced $SO_2$ levels, especially in the summer which involves more than 55% of the highest $SO_2$ concentrations (>20µg/m$^3$). Local sources of $SO_2$ and UFPs include not only the large petrochemical and industrial area of Fos-Berre, located 40 km northwest, but also the shipping traffic related to these activities in the gulf of Fos. Fresher emission of $SO_2$ can originate from the port of Marseille, located 3 km away from the site. During summertime, the ships traffic increases by 25% (4319 against 3262 for the 2017-2018 period) partly because of the enhanced numbers of passenger ferries and travel cruises during the holiday season.

Figure 9 shows the diurnal trend of cumulative number of ship movements for the summer 2017. This number was differentiated according to the type of movement and the basin location (the exact geographic positions are reported in the Supplement in Figure S1). As can be seen by the diurnal profile of the ship movements in the harbour, two maxima were observed: the first at 05:00 UTC due to ship arrivals and the second at 17:00 UTC due to ship departures. The highest $SO_2$ peak might be related to the ship arrivals increase; however, El Haddad et al. (2011a, 2013) assigned similar morning $SO_2$ plumes to industrial activity from Fos-sur-mer in summer. Thus, the first $SO_2$ increase could arise from a combination of contribution from ships arrival and industrial air masses, whereas the second peak could be mostly linked to the late afternoon boats departure. $SO_4^{2-}$ concentrations increased during the day and could be also partly affiliated to the direct influence of shipping/industrial activity to the monitoring site.

To further investigate the large $SO_4^{2-}$ concentrations in summer and their origin at a broader scale, a cluster analysis was carried out on the air masses reaching the site. From the analysis performed on the 72h-backtrajectories, 3 distinct clusters were assessed from the total spatial variance (TSV) variation. Cluster 1 (Mediterranean origin) is related to air masses that circulate through western Mediterranean basin before arriving at MRS-LCP site. Cluster 2 (sea breeze) corresponds to an initial low mistral blowing along the Rhone Valley seaward that returns to land the next day. Cluster 3, similarly to cluster 2, is representative of Mistral wind from the Rhone Valley but with higher speeds, as recorded at MRS-LCP (average of 1.01 m $s^{-1}$ against 0.60 and 0.72 m $s^{-1}$ for cluster 1 and 2, respectively). Mean calculated trajectories are displayed in Figure S10. Cluster 3 associated to low pollutant levels won't be investigated in this study as its frequency is low (19%). In comparison, Mediterranean and sea breezes air masses account for 43% and 38%, respectively. While the clustering analysis clearly identifies the Mediterranean long range trajectories, Figure S11 shows that they still get mixed with the sea breeze when they approach the shore (as indicated by the wind sector 190°-270° characteristic of the sea breeze, and the by the sharp $SO_2$ peaks included in the Mediterranean regime periods in pink). Still, discernible differences in the particle distribution and in the chemical composition are observed, as shown by Figure 10a and Figure 10b. Sea breeze cluster had a pronounced nucleation mode and Aitken mode (>10.25 nm according to SMPS measurements) (Figure 10a). Mediterranean origin cluster has a broaden size distribution with combined Aitken and accumulation modes. Mallet et al. (2019) found similar results at Lampedusa site for air masses passing over the western Mediterranean. Particles from the sea breeze cluster might be smaller than Mediterranean origin cluster with the likelihood of fresher emission from the nearby shipping/industrial sources. Moreover the Box plots of $SO_4^{2-}$, $SO_2$ and $N_{2(10\text{-}20\ nm)}$ concentrations related to cluster 1 and 2 reveal that slightly higher $SO_2$

(3.1 vs 2.6 µg $m^{-3}$) and $N_{2(10\text{-}20\ nm)}$ (7855 vs 5740 $cm^{-3}$) concentrations are encountered with cluster 2, while higher $SO_4^{2-}$ concentrations are observed with cluster 1 (3.2 µg $m^{-3}$ against 1.9 µg $m^{-3}$ for sea breeze).

In an effort to further investigate the characteristics of the sulfate constituents, the total $SO_4^{2-}$ was first tentatively deconvolved into ammonium sulfate, organosulfate and MSA (methanesulfonic acid) following the methodology developed by Chen et al. (2019) and based on $HSO_3^+$ (f81) and $H_2SO_4^+$ (f98) ion fractions from AMS measurements. The resulting $fH_2SO_4^+$ vs $fHSO_3^+$ data points are displayed in Figure 12 and are color-coded according to the different air masses (Sea breeze and Mediterranean origin). The results show that ammonium sulfate was the dominant source for ToF-ACSM sulfate signals during this period. This was confirmed by the $NH_{4\ measured}/NH_{4\ predicted}$ ratio close to 1 (Figure S4), showing that $SO_4^{2-}$ was always fully neutralized by $NH_4^+$ at MRS-LCP site. It is hypothesised that even local emissions have enough time to mix with ammonia from urban environment to fully neutralize before reaching the station: considering a minimum distance between the harbour and MRS-LCP of 3 km and an average sea breeze wind speed between 1.05 m $s^{-1}$ and 2.7 m $s^{-1}$ (measured from a harbour meteo station under the same sea breeze blowing direction) it takes at least around 20-45 minutes for a plume to reach the station. This agrees with Celik et al. (2020) who observed that shipping plumes older than 40 min were mostly neutralized with the ambient $NH_3$ from a cleaner environment.

In order to trace the geographical origin of $SO_4^{2-}$ from MRS-LCP at a regional scale, CWT analysis were performed and the results are shown in Figure 11. A weighting function has been implemented to avoid artefacts linked to high concentrations with low number of trajectories passing through a particular cell. The aim is to avoid local sulfate influence from the transport model. Following Waked et al. (2014) recommendations a discrete function based on back trajectory density ($\log_{10}(n+1)$) was applied (using the ZeFir tool on Igor). CWT on $SO_4^{2-}$ from cluster 1 (Figure 11a) pointed out the combined influence of low Mistral advection from the Rhone Valley and the switch into South-Westerly thermic breeze, in agreement with the expectations. CWT for cluster 2 (Figure 11b) exhibited long-range transport, with $SO_4^{2-}$ concentrations associated to the south and western Mediterranean air mass circulation. In the basis of this, several sources can be related to sulfate emissions in the Mediterranean Sea such as shipping activity, marine biogenic or crustal origin (Becagli et al., 2012).

## 4 Summary and conclusions

The chemical composition of submicron aerosols was monitored in real time between 1 February 2017 and 13 April 2018 at an urban background site of the Mediterranean city of Marseille. Measurements were carried out with a ToF-ACSM associated with a suite of collocated instruments including an aethalometer, an ultrafine particle monitor, a SMPS and monitors for regulated pollutants (PM, $NO_x$, $O_3$, $SO_2$).

The reconstituted $PM_1$ mass (ACSM measurements + BC) was cross validated through several comparisons with external parameters. ACSM+BC concentrations were found to be in good agreement with estimated mass concentrations from SMPS ($R^2$=0.81 and slope=1.02) and $PM_1$ concentrations from FIDAS ($R^2$=0.89 and slope=0.96).

OA was the most abundant specie of submicron aerosol, with an annual average of 49.7% and the carbonaceous fraction was dominant for every season (66.8%) and especially during cold months. BC contributes largely to this fraction (17.1% of total submicron aerosol) and is mainly dominated by fossil fuel emissions as determined with the AE-33 aethalometer model. BC from wood burning emissions showed higher contribution during winter and very low contribution in summer, as expected.

The organic nitrate contribution was evaluated using the $NO_2^+/NO^+$ ratio method and gave reasonable results in separating $NO_{3,Org}$ and $NO_{3,Inorg}$ concentrations. Some uncertainties still remain as $R_{ON}$ was set to a fixed value and could slightly vary according to the VOC precursors which lead to particle organic nitrate formation. Also, low nitrate signal provide instable $R_{obs}$ and could enhance the uncertainty of the estimation. The $NO_{3,Org}$ fraction was 20±7% for the total nitrate (representing 10.2% on average for the $PM_1$ concentrations) during the entire period and did not significantly contribute to enhanced polluted events with high $PM_1$ concentrations. However organic nitrate contribution to total OA could be estimated to significant values with maximum during springtime (18-28%).

Particle number concentration was successfully segregated into two components ($N_1$ and $N_2$) by using the minimum slope found in the N vs $BC_{FF}$ plot, with $BC_{FF}$ accounting for primary particles. $N_1$ was attributed to fresh primary traffic emissions and $N_2$ to secondary particles. The secondary $N_2$ fraction was predominant with number concentrations 1.02 to 1.7 times higher than $N_1$. While $N_1$ showed clear maxima during morning and evening traffic rush hours, $N_2$ was either attributed to dilution/cooling and mixing of vehicular exhausts in the atmosphere during cold seasons or to photooxidation products of gaseous precursors during hot seasons. These results revealed the importance of secondary particle formation and contrasted seasonal sources of UFPs number.

$PM_1$ pollution events were determined according to the daily concentrations exceeded WHO recommendations. To illustrate their differences in chemical composition, meteorological dynamic and geographical origins two events (23-24 December 2017 and 22-25 February 2018) were carefully examined. $BC/SO_4^{2-}$ ratio, non-parametric wind regressions and back trajectories provided important information to discriminate local and long-range transport contributions. The local contribution during exceedance days is attributed to an increase of biomass burning emissions with domestic heating and green wastes burning, cumulated with intense traffic. In those situations, the OA and BC concentrations strongly increase at night in the space of few minutes when the nocturnal land breeze set up and the boundary layer height decreases. The long-range pollution case, led to high increase of secondary aerosol, more precisely ammonium nitrate and oxygenated OA, transported from the central Europe and notably Pô Valley to the city. The investigation of these two episodes highlighted that local influences are mainly responsible of continuous background pollution and its mix with long-range transport events can trigger to situation with high exceedance levels of fine particles. This elevated pollution occurred mostly in winter and early spring with favourable wind conditions. Even if no exceedance day was found during the summer season, the chemical composition is slightly different, with higher sulfate contributions and intense UFPs plumes (mostly between 10 and 20 nm) associated with $SO_2$ are advected on site. Air mass clustering has been performed to explain the observed differences in aerosol composition and highlighted the presence of local and regional emissions of shipping activity, mixed with industrial plumes in Marseille. This configuration is expected in a coastal city with a consequent harbour and in the vicinity of an industrial area. First sea breeze

blowing results in a local advection of $SO_2$, UFPs and $SO_4^{2-}$ from the industrial/shipping plumes. Then regional Mediterranean air masses can bring higher sulfate concentrations and larger particles from aged shipping plumes, probably mixed with other sources such as marine biogenic or crustal. For both cases it has been shown that sulfate was completely and rapidly neutralized suggesting that ammonium sulfate predominates in the Mediterranean Sea.

In conclusion, the supersite MRS-LCP successfully recorded long-term observations and seasonality of fine particles. The long-term real-time monitoring in MRS-LCP showed a great potential and will supply direct information to public authorities and citizens. It may provide better understanding of pollution episodes and more effective control of local mitigation within the context of French Atmosphere Protection Plan.

***Data availability.*** Data are available upon request to the contact author Benjamin Chazeau (benjamin.chazeau@univ-amu.fr).

***Author contributions.*** NM designed the research. BC, GG and BM contributed to the measurements. BC performed the analysis and wrote the paper. NM, BD, BT and HW reviewed and all authors commented on the paper.

***Competing interests.*** The authors declare they have no conflict of interest.

***Acknowledgements.*** This work is supported by AtmoSud, ANRT, the PACA Region and the French ministry of Environment. BC also acknowledges V Crenn and ADDAIR for the ToF-ACSM support, D Piga for the supply of shipping data and J-E Petit for developing Zefir tool.

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

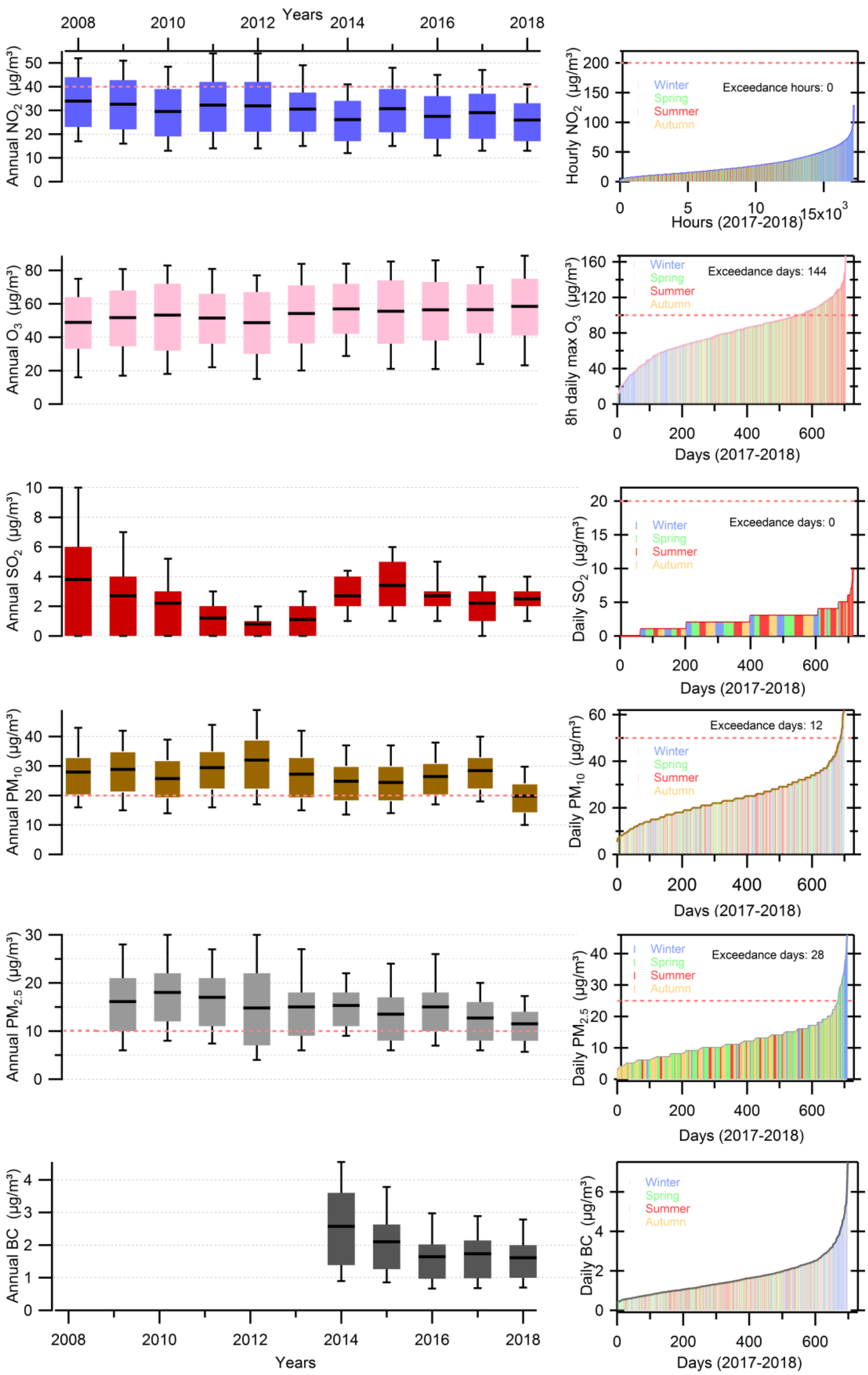

**Figure 1: Graphs at left show annual average concentrations of the following pollutants during the last 11 years in Marseille-Longchamp: Nitrogen dioxide (Blue), Ozone (pink), sulphur dioxide (red), $PM_{10}$ (brown), $PM_{2.5}$ (grey) and Black carbon (black). The red dotted thresholds correspond to the annual concentrations recommended by WHO (World Health Organisation) for Nitrogen dioxide, $PM_{10}$ and $PM_{2.5}$. Graphs at right show data sorted by increasing daily values together with the number of days in excess according to the WHO recommendations (again the red dotted lines) for the years 2017 and 2018. For ozone, it corresponds to the daily maximum 8h average. Sticks for daily values are color-coded depending on the seasons (winter = blue; spring = green; summer = red; autumn = orange).**

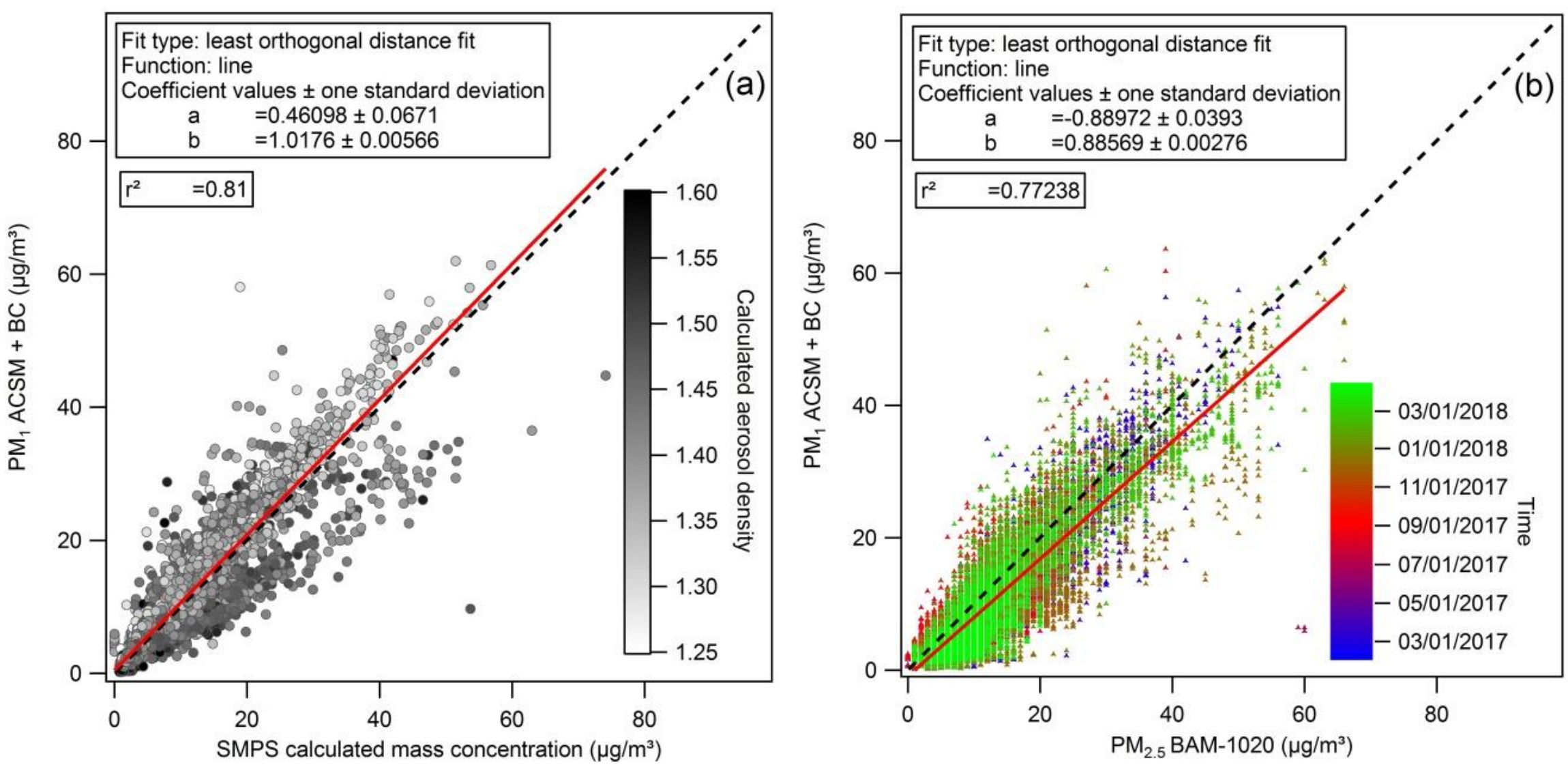

**Figure 2: Reconstructed $PM_1$ (ACSM + BC) vs $PM_1$ calculated from SMPS measurements, with the color-coded density evaluated from the $PM_1$ chemical composition (a) and reconstructed $PM_1$ vs PM2.5 measurements by BAM-1020, coloured according to the sampling time (mm/dd/yyyy) (b). Red lines correspond to orthogonal distance fits and black dashed lines to 1:1 lines. $R^2$ are determined with least square fits.**

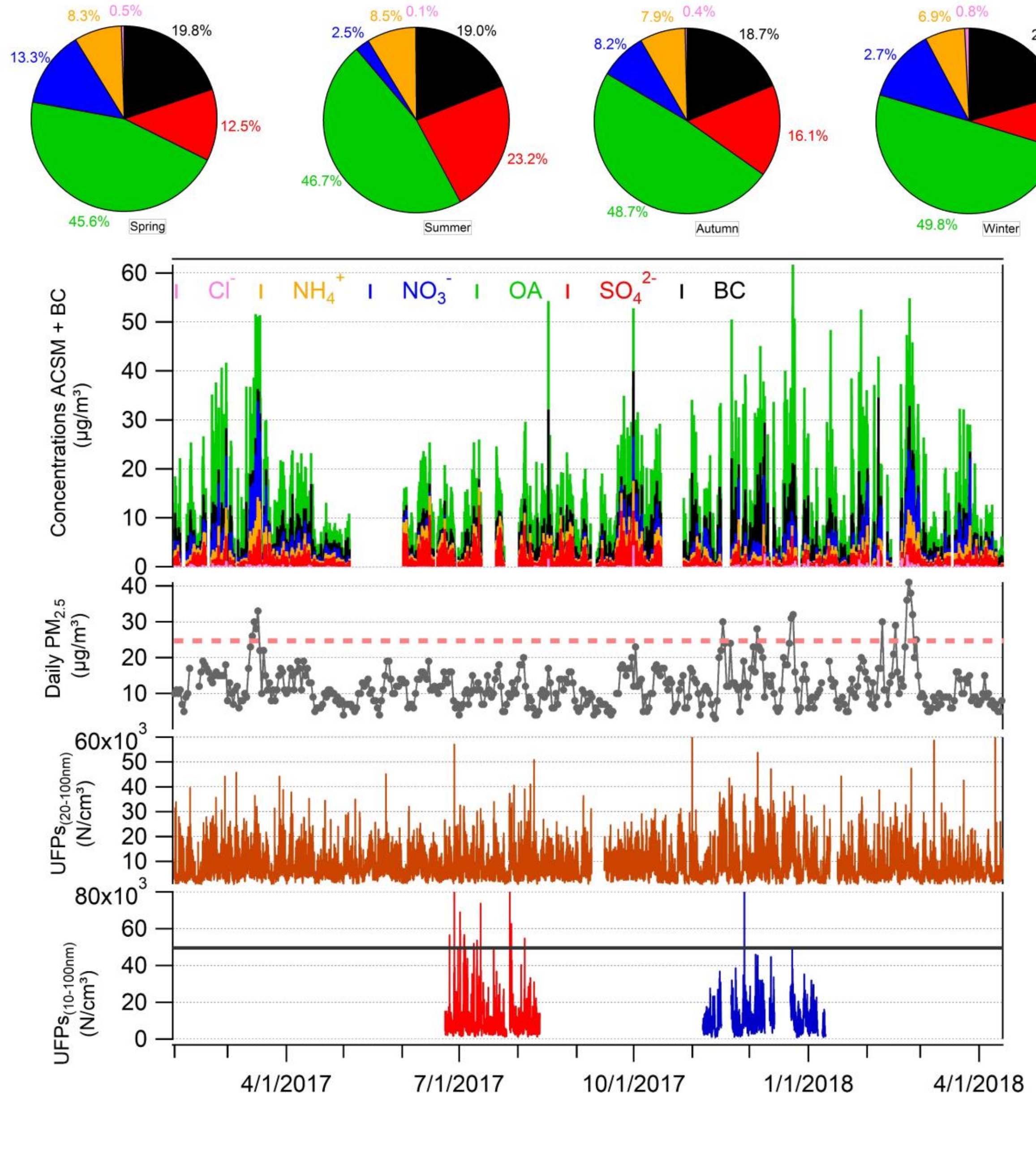

1030 **Figure 3: Time series of 1h-$PM_1$ chemical species ($Cl^-$, OA, $NH_4^+$, $NO_3^-$, $SO_4^{2-}$ and BC), daily-$PM_{2.5}$, total UFPs (20-100nm) measured with 3031 and total UFPs (10-100nm) measured with SMPS GRIMM (summer period in red and winter period in blue) from 1 February 2017 to 13 April 2018. The average particle number concentrations were higher in winter however sharp events exceeding 50 000 particles $cm^{-3}$ (black line) were more abundant in summer period. This feature is traceable for the 10-100 nm size range measured with the SMPS GRIMM. The pink dotted thresholds correspond to the daily $PM_{2.5}$ concentrations recommended by WHO. Pie charts of averaged $PM_1$ components contribution for each season are showed on the upper panel.**

| | | SPRING (March-April-May) | | SUMMER (June-July-August) | | AUTUMN (September-October-November) | | WINTER (December-January-February) | |
|---|---|---|---|---|---|---|---|---|---|
| | Unit | Average | SD | Average | SD | Average | SD | Average | SD |
| OA | µg $m^{-3}$ | 3.86 | 3.37 | 4.55 | 2.66 | 5.07 | 3.78 | 6.17 | 5.69 |
| BC | µg $m^{-3}$ | 1.30 | 1.17 | 1.49 | 1.16 | 1.90 | 1.74 | 2.12 | 2.27 |
| $SO_4^{2-}$ | µg $m^{-3}$ | 1.06 | 0.84 | 2.26 | 1.66 | 1.57 | 1.50 | 1.12 | 0.99 |
| $NO_3^-$ | µg $m^{-3}$ | 1.13 | 2.08 | 0.24 | 0.23 | 0.83 | 1.09 | 1.58 | 2.13 |
| $NH_4^+$ | µg $m^{-3}$ | 0.70 | 0.85 | 0.83 | 0.59 | 0.77 | 0.77 | 0.86 | 0.93 |
| $Cl^-$ | µg $m^{-3}$ | 0.04 | 0.08 | 0.01 | 0.04 | 0.04 | 0.11 | 0.09 | 0.18 |
| $PM_1$ | µg $m^{-3}$ | 8.09 | 6.99 | 9.39 | 4.77 | 10.2 | 7.26 | 11.9 | 9.79 |
| $BC_{WB}$ | µg $m^{-3}$ | 0.19 | 0.25 | 0.11 | 0.14 | 0.24 | 0.46 | 0.59 | 0.89 |
| $BC_{FF}$ | µg $m^{-3}$ | 1.10 | 1.04 | 1.38 | 1.10 | 1.66 | 1.53 | 1.52 | 1.78 |
| $NO_{3,Org}$ | µg $m^{-3}$ | 0.22 | 0.32 | 0.09 | 0.08 | 0.18 | 0.17 | 0.26 | 0.26 |
| $NO_{3,Inorg}$ | µg $m^{-3}$ | 0.91 | 1.80 | 0.15 | 0.18 | 0.64 | 0.94 | 1.31 | 1.95 |
| $N_{1\ (20\text{-}100nm)}$ | N $cm^{-3}$ | 3200 | 2531 | 2756 | 1736 | 3804 | 3212 | 4268 | 3916 |
| $N_{2\ (20\text{-}100nm)}$ | N $cm^{-3}$ | 4339 | 3755 | 4777 | 4552 | 4301 | 3958 | 4332 | 3641 |

**Table 1: Seasonal average concentrations and associated standard deviation (SD) of main chemical species of submicron aerosol for the study period. Average concentrations and SD of BC, nitrate, and particles number (20-100 nm) components are also represented.**

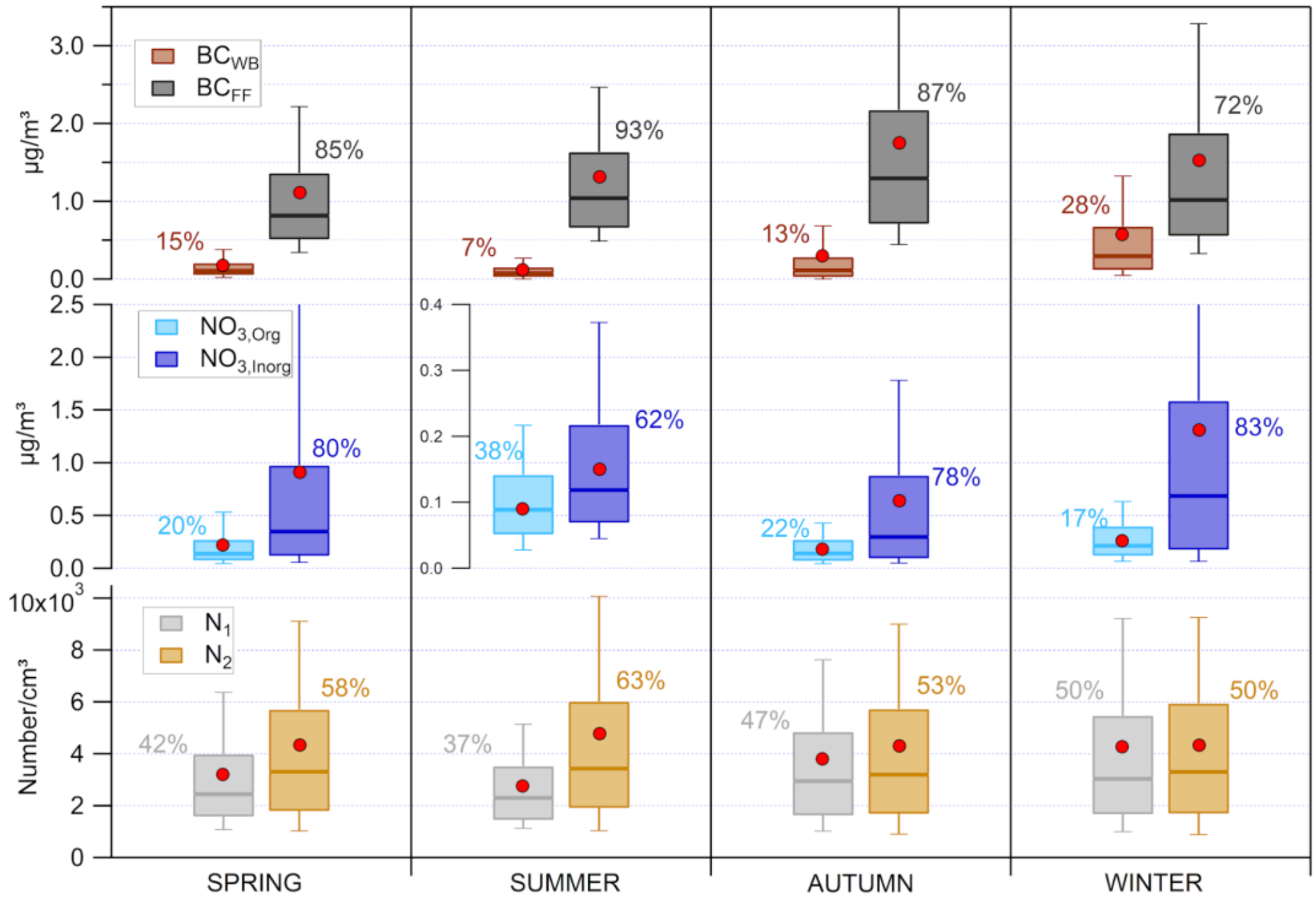

**Figure 4: Seasonal concentrations for the different fractions of BC ($BC_{WB}$ and $BC_{FF}$), $NO_3^-$ ($NO_{3,Org}$, $NO_{3,Inorg}$), and UFPs number between 20 and 100nm ($N_1$, $N_2$) represented as box plots. The band inside the box is the median (50th percentile), the bottom and top of the box represent the lower and upper quartiles respectively (the 25th and the 75th percentile). The ends of the whiskers denote here the 10th and 90th percentile. The red dots refer to the mean of each component.**

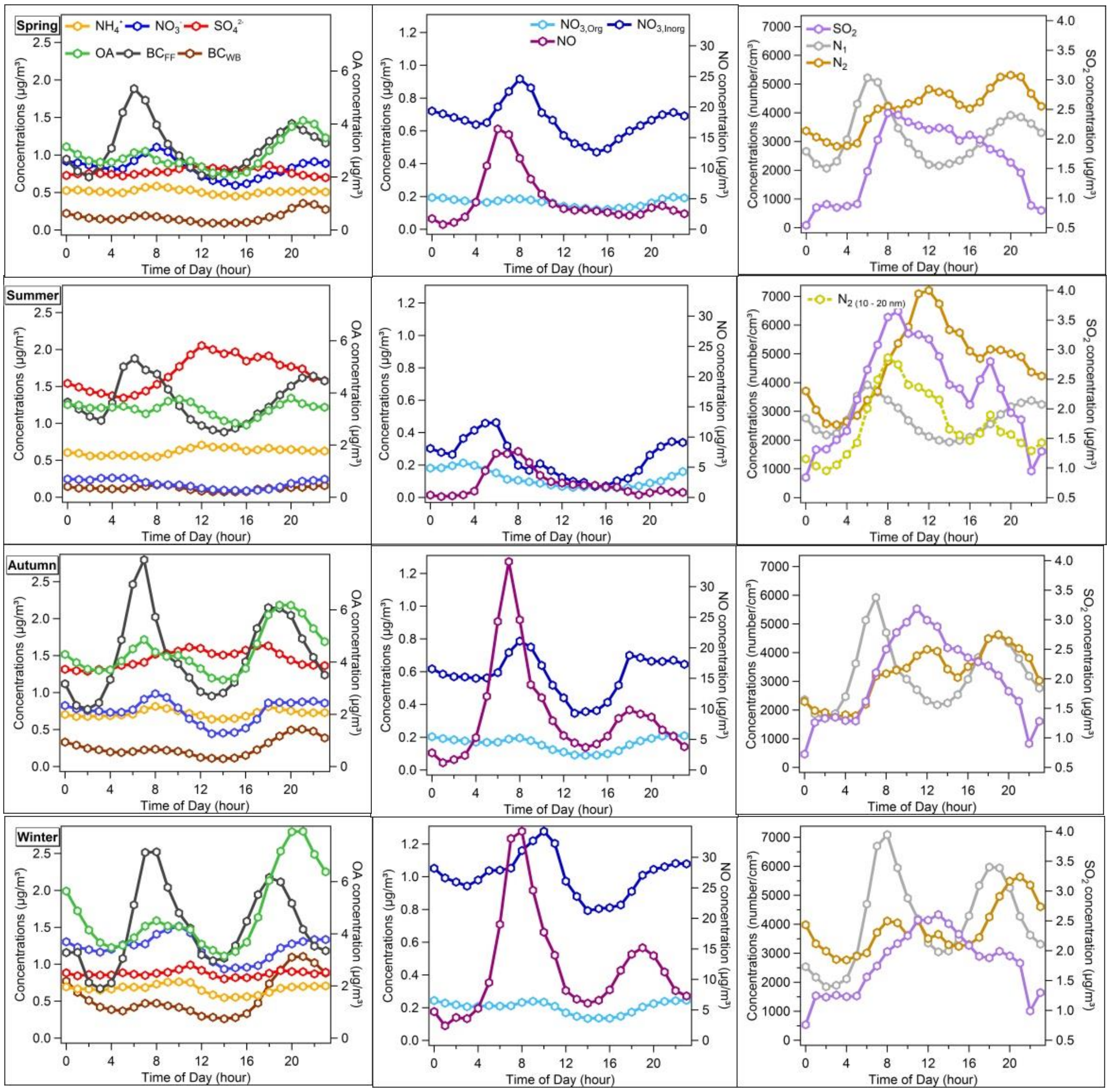

**Figure 5: Seasonal diurnal profiles of $PM_1$ species (left part), nitrate components ($NO_{3,Org}$, $NO_{3,Inorg}$) and NO (middle part), and UFPs components ($N_1$, $N_2$) and $SO_2$ (right part). For summer, $NO_{3,Org}$ profile correspond to June 2017 only, and secondary number of particles $N_2$ was also calculated for N between 10 and 20 nm from SMPS GRIMM measurements during the 23 June-12 August 2017 period (yellow dashed-line).**

| | 2008 | 2009 | 2010 | 2011 | 2012 | 2013 | 2014 | 2015 | 2016 | 2017 | 2018 |
|---|---|---|---|---|---|---|---|---|---|---|---|
| **Winter** | - | 12 | 19 | 29 | 31 | 35 | 4 | 17 | 18 | 11 | 7 |
| **Spring** | - | 10 | 16 | 11 | 16 | 5 | 11 | 8 | 3 | 4 | 0 |
| **Summer** | 0 | 13 | 19 | 3 | 1 | 4 | 1 | 0 | 2 | 0 | - |
| **Autumn** | - | 15 | 7 | 14 | 3 | 3 | 4 | 6 | 5 | 1 | - |
| **Year** | - | 50 | 61 | 57 | 51 | 47 | 20 | 31 | 28 | 16 | - |

**Table 2: Number of days exceeding 25 µg m$^{-3}$ for $PM_{2.5}$ concentrations (WHO recommendation) since they are measured in MRS-LCP. Results are presented by season and for the total years.**

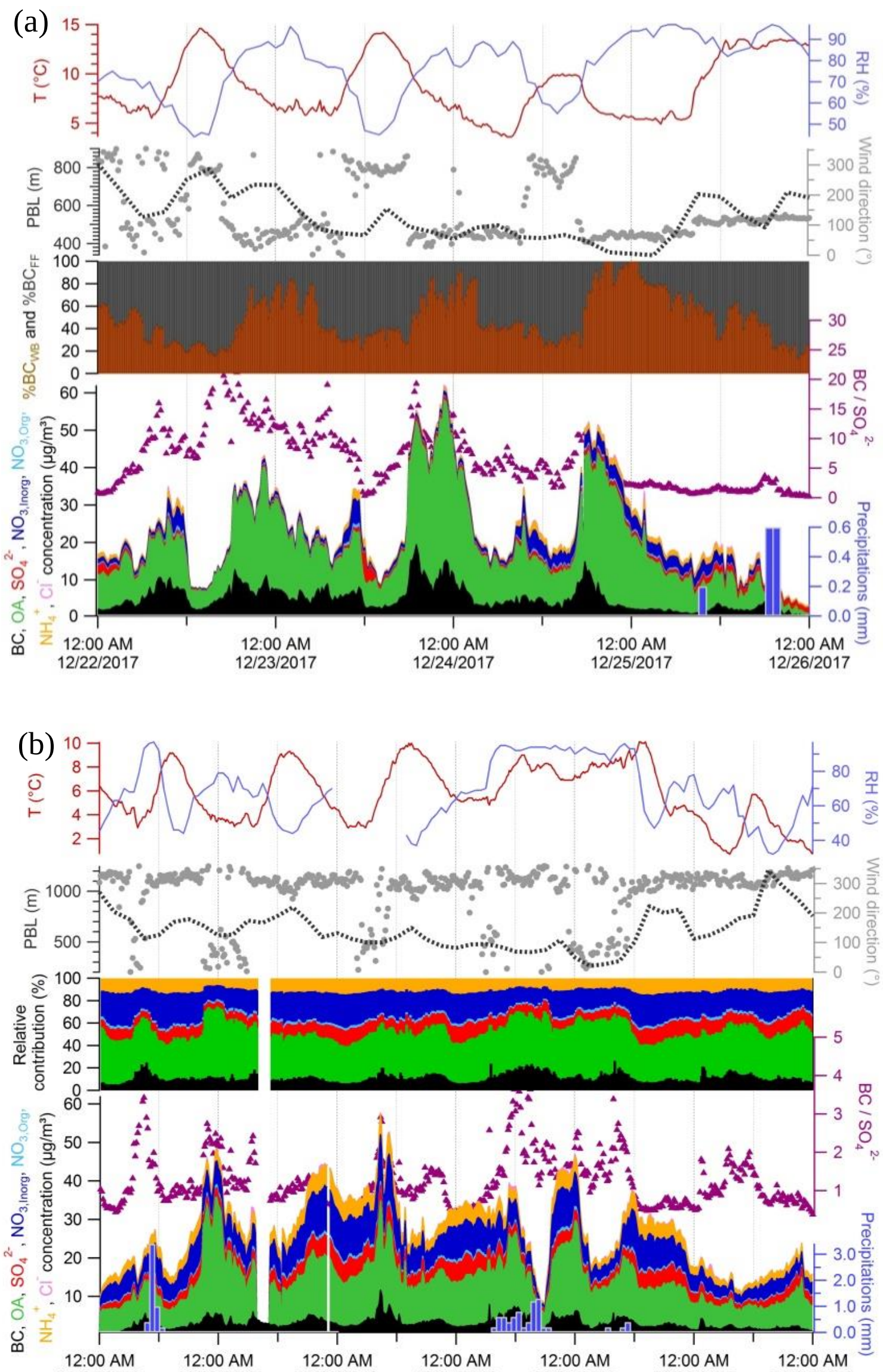

**Figure 6: Time series of meteorological conditions and chemical composition during the two polluted episodes. Graph (a) represents for the local Christmas event (22-26 December 2017) the chemical composition of submicron particles ($Cl^-$, $NH_4^+$, $NO_{3,Org}$, $NO_{3,Inorg}$, $SO_4^{2-}$, OA and BC); the relative contribution of $BC_{FF}$ and $BC_{WB}$; the $BC/SO_4^{2-}$ ratio; and the meteorological data (temperature, relative humidity, wind direction, planetary boundary layer and precipitations). Graph (b) shows for the long-range event (21-27**

**February 2018) the chemical composition of submicron particles; the relative contribution of each specie; the BC/$SO_4^{2-}$ ratio; and the meteorological data.**

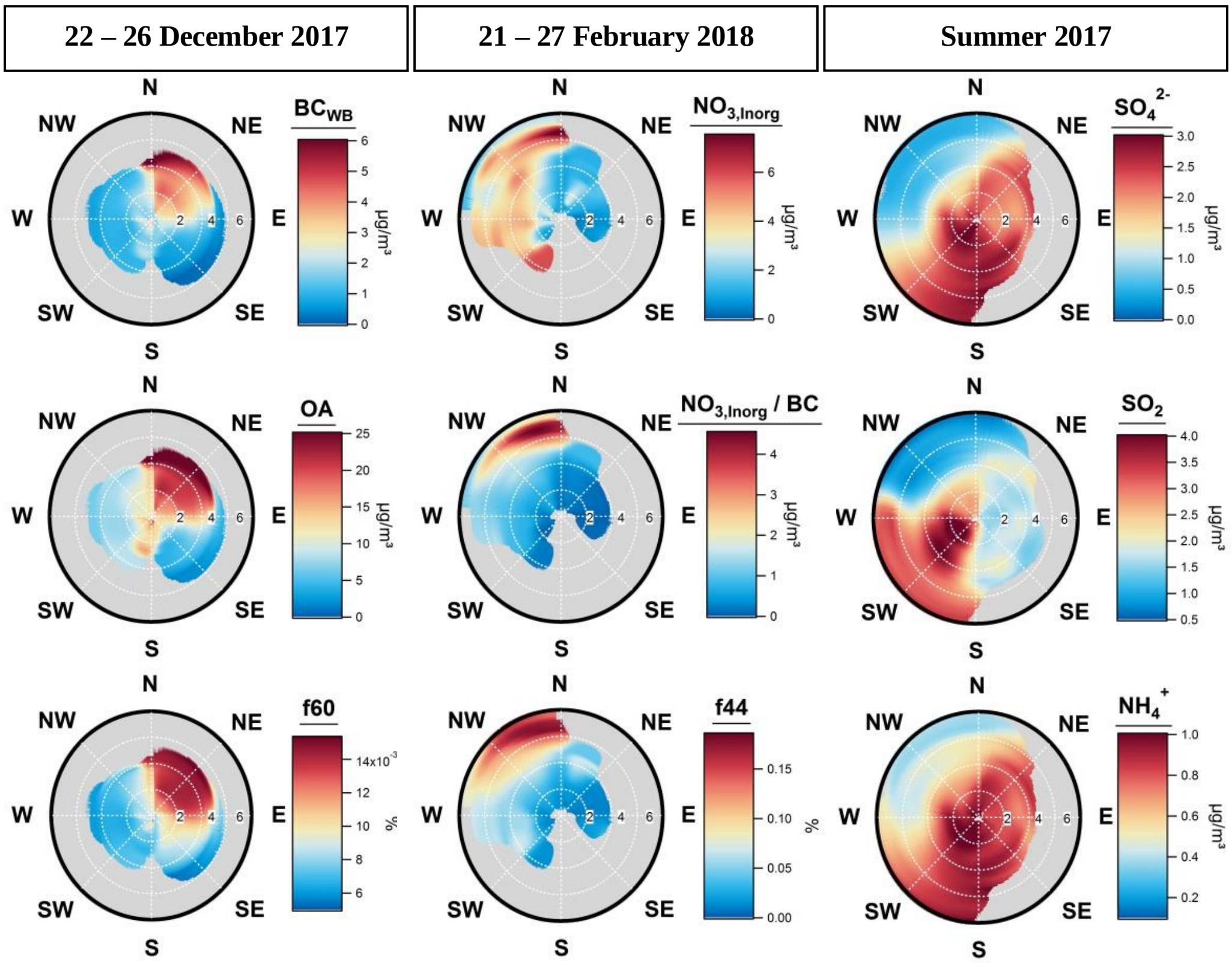

**Figure 7: NWR plots for the local Christmas event, SWIM-2 plots for the long-range episode and NWR plots for the summer period. $BC_{WB}$ concentrations, OA concentrations and fraction of the m/z 60 ACSM organic signal are represented for the local Christmas event from 22 to 26 December 2017. $NO_{3,Inorg}$ concentrations, $NO_{3,Inorg}$ normalized by BC and fraction of the m/z 44 ACSM organic signal are displayed for the long-range influence from 21 to 27 February 2018. $SO_4^{2-}$, $SO_2$ and $NH_4^+$ concentrations are represented for the summer 2017. Radial and tangential axes show respectively the wind speed (m.s$^{-1}$) and the wind direction (°).**

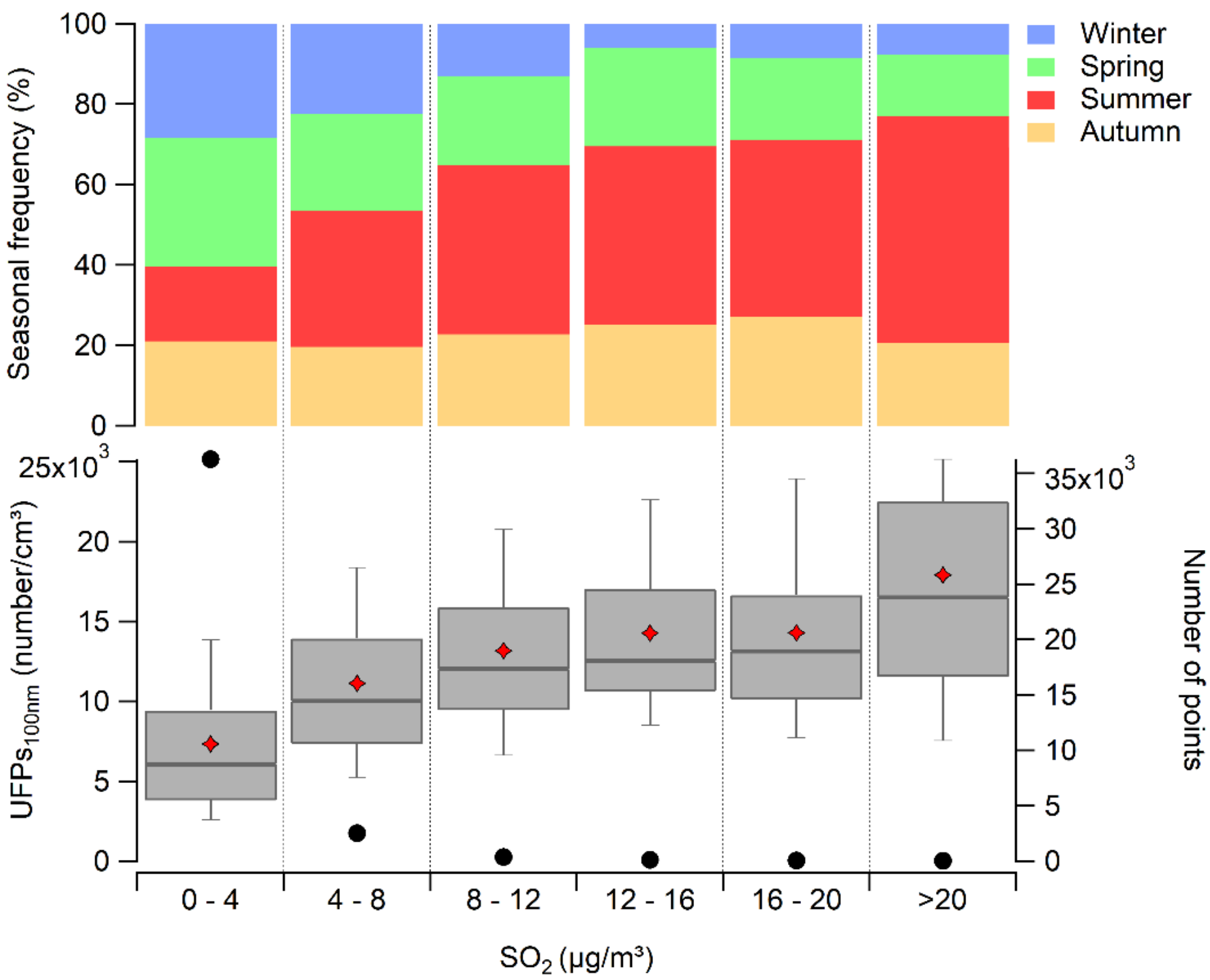

**Figure 8: Box plots of UFPs$_{<100nm}$ number for different $SO_2$ concentrations classes (bottom) and seasonal occurrence frequency for each class (top). The red diamonds are the mean, the bands inside the box are the median, the bottom and top of the box represent the lower and upper quartiles respectively and the ends of the whiskers show the 10$^{th}$ and 90$^{th}$ percentile. The black circles denote the number of points encountered in each bin.**

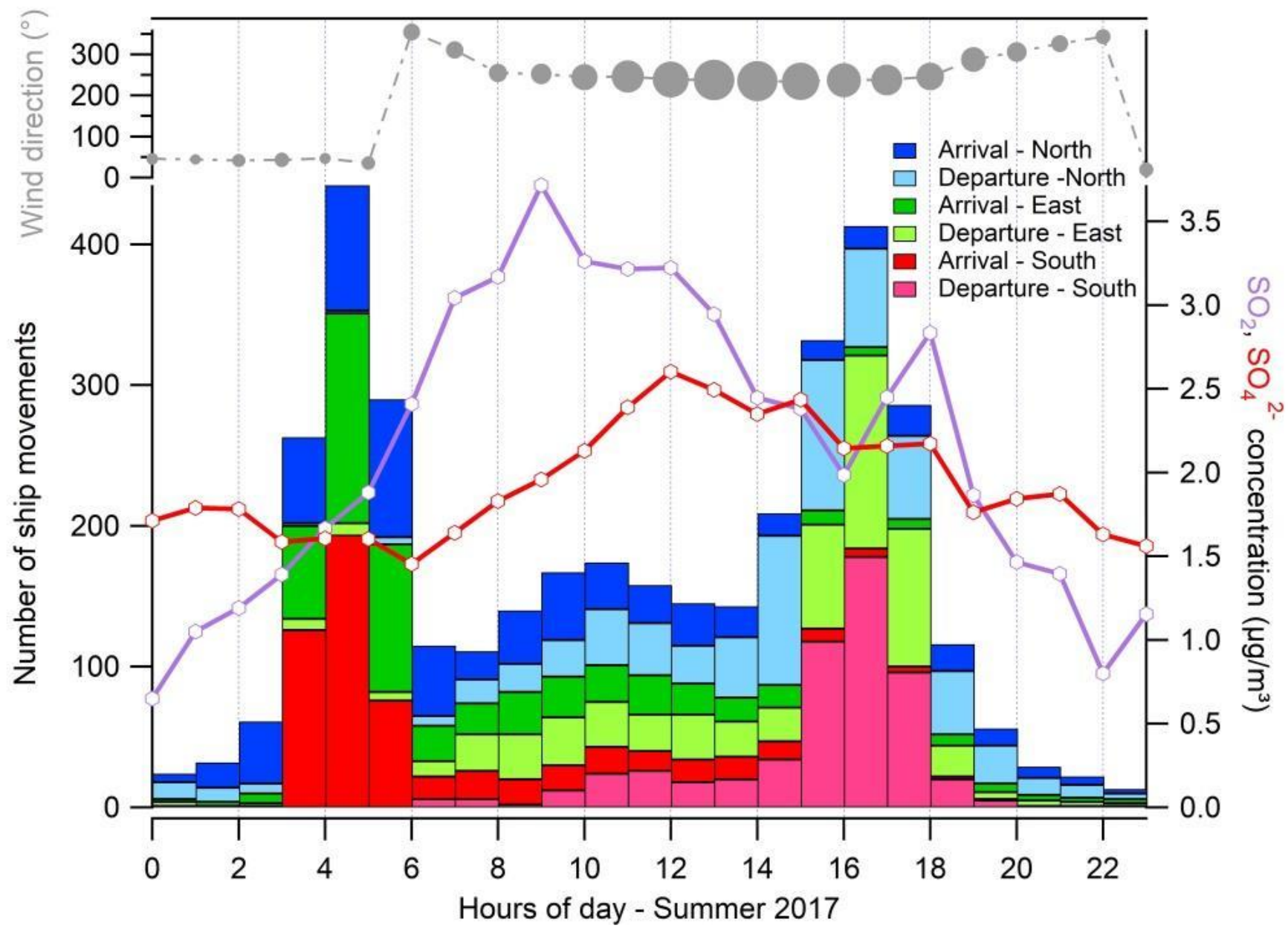

1090 **Figure 9: Diurnal profile of cumulative number of ship movements for the summer 2017 period, coloured according to the basin location (blue for north, green for east, red for south) and the type of movement (dark-coloured for arrival, light-coloured for departure). Diurnal profiles of $SO_2$ in purple, $SO_4^{2-}$ in red and wind direction in grey (the size of dots is proportional to the wind speed intensity) are also represented. The data points linked to low speed conditions (<0.5 m $s^{-1}$) were filtered out from the analysis.**

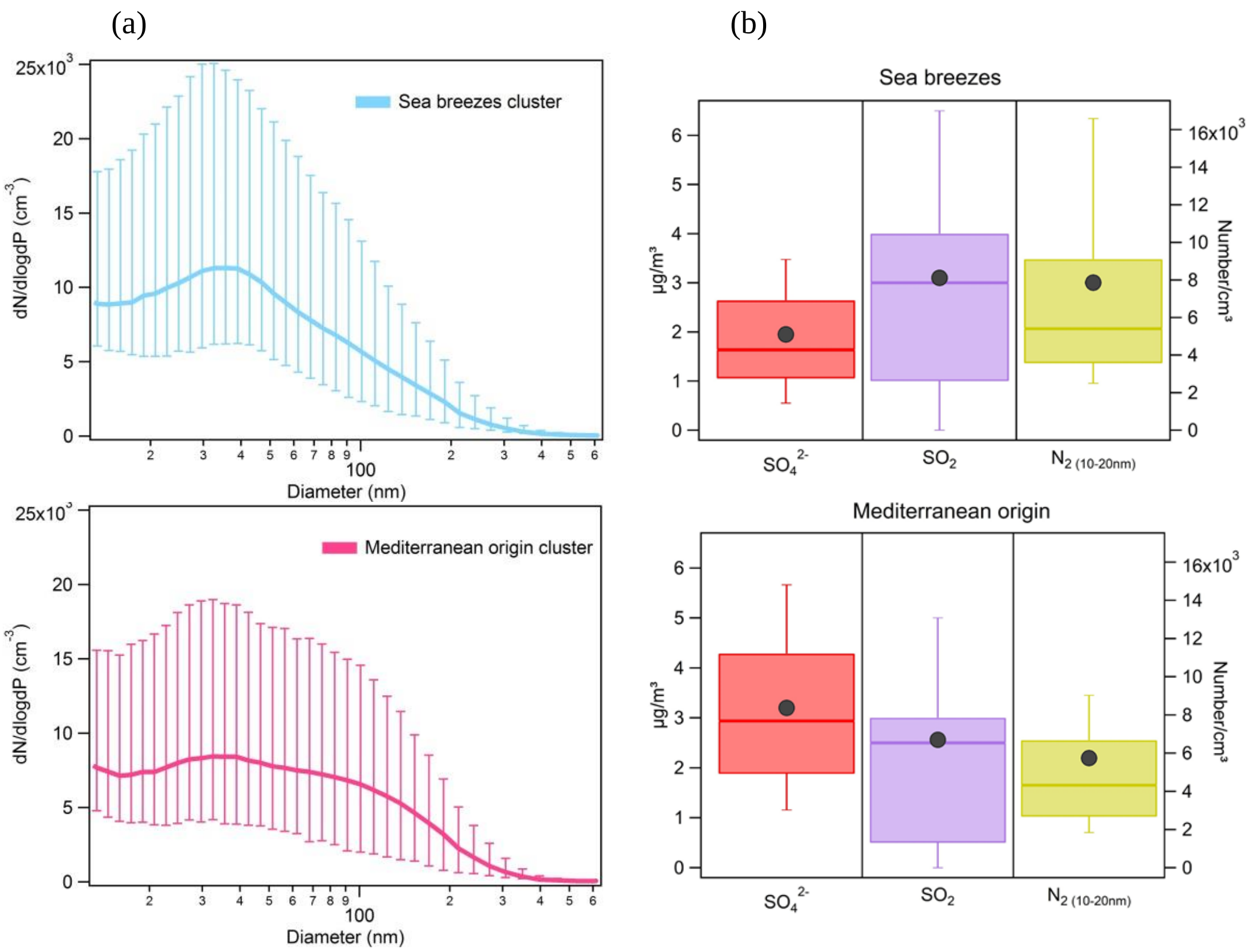

**Figure 10: (a) The number size distribution between 10 and 600 nm from SMPS GRIMM, coloured by average for different clusters (sea breezes/cluster 2 in light blue and Mediterranean origin/cluster 1 in pink) during the summer 2017 period. (b) Box plots for $SO_4^{2-}$, $SO_2$ and $N_{2\,(10\text{-}20\,nm)}$ concentrations represented for the two air mass origins.**

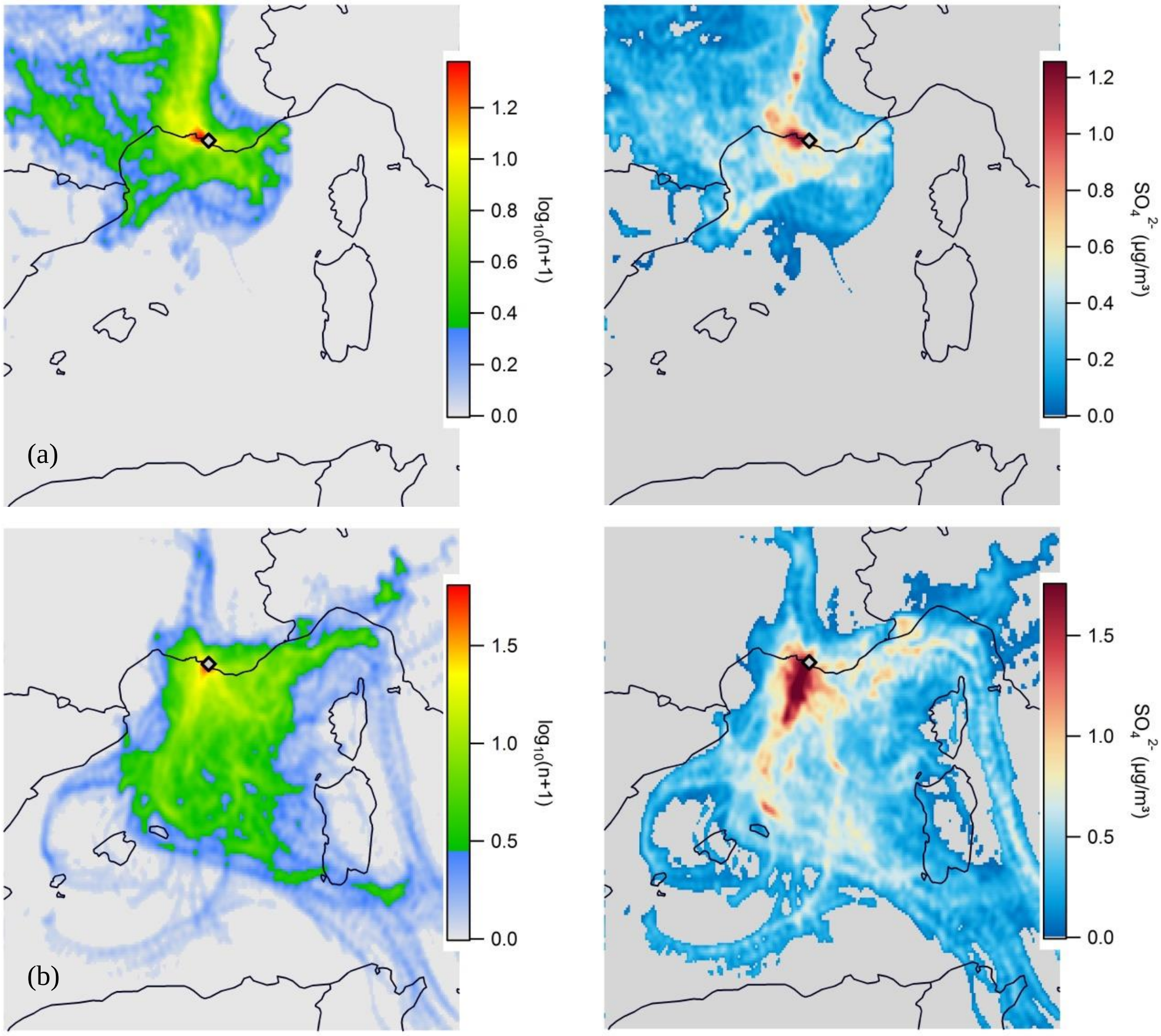

1105 **Figure 11: The left panel shows ZeFir trajectory density for 72h-backtrajectories generated every hour at MRS-LCP station for (a) the sea breezes cluster and (b) the Mediterranean origin cluster. The $log_{10}(n+1)$ colour scale corresponds to the occurrence of backtrajectory endpoints which drop into a particular cell. CWT maps for $SO_4^{2-}$ (in µg $m^{-3}$) are displayed on the right panel for the two clusters.**

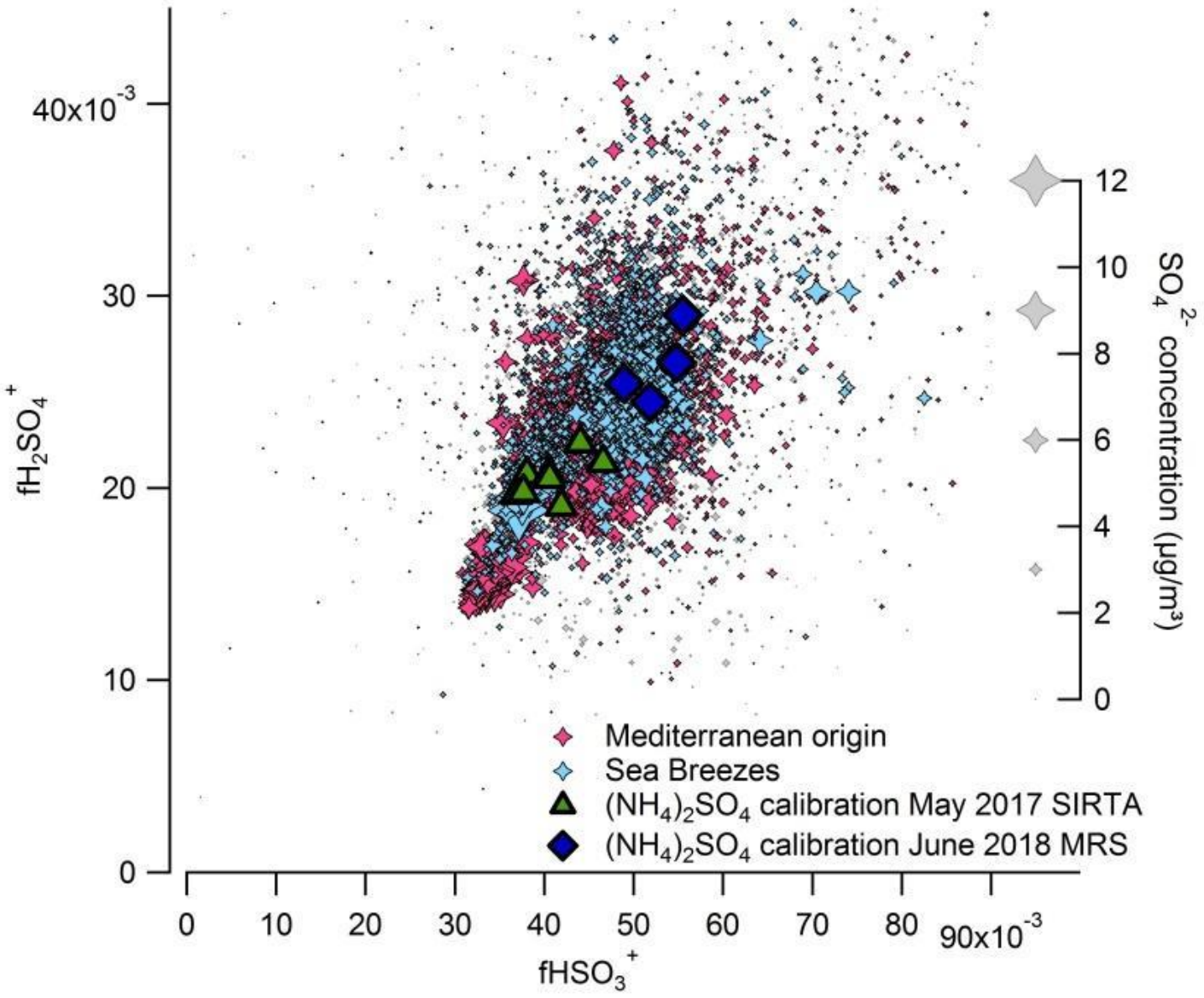

1110 **Figure 12: $fH_2SO_4^+$ vs $fHSO_3^+$ for summer 2017 ACSM measurements at MRS-LCP site. Blue diamonds and green triangles are values obtained during the corresponding ammonium sulfate standard calibrations. The pink and light blue markers are data obtained for the Mediterranean origin and sea breezes clusters, respectively. Marker sizes are proportional to $SO_4^{2-}$ mass concentration in order to minimize the ratio of signals closed to detection limit.**