# Peer review of "Measurement report: Fourteen months of real-time characterisation of the submicronic aerosol and its atmospheric dynamic at the Marseille-Longchamp supersite"

_Atmospheric Chemistry and Physics, 2020_

## Referee Comment (RC1) · Anonymous Referee #1 · 25 Nov 2020

Review

Chazeau et al. describe a 14-months measurement campaign at the urban station of Marseille-Longchamp supersite (Marseille, France) from February 2017 to April 2018. The paper focuses on the analysis of data from ToF-ACSM and aethalometer measurements. Besides, it is important to note that the station is also used by the local air quality agency providing long-term measurements of standard air quality variables

(NOx, O3, SO2; BC, PM10, and PM2.5), which are partly included in the manuscript. The discussion begins with a mass closure analysis using collocated instrumentations, following by a description of the seasonal and the diurnal profiles of the PM2.5 chemical components. Two case-study events corresponding to periods when the total PM mass exceeded the WHO recommendation of 25 $\mu$g/m3 over 24 h were considered and described in more detail (Christmas 2017 and 4 consecutive days in February 2018). Finally, the results section ends with a discussion on the influence of ship and industry emissions on the local sulfate mass concentration. Although the manuscript responds to the need for a +1 year continuous measurements at a high time-resolution to better characterize the different factors and sources influencing local air quality, the results are presented rather as a descriptive report than as an attempt to answer a well-identified scientific question or to focus on a specific topic. This lack of a central theme makes it difficult to read and to catch the link between the different sections and the sub-sections. A direct consequence is that sometimes, I had the impression that the authors have lost their focus and started to describe results that are not directly related to the measurements made at the Marseille-longchamp supersite (for example, the discussion about BC source at the Kaddouz site) or to change the subject before returning to it (for example, the seasonal variability section starts and ends on BC source estimation, with in-between results from the splitting of the ToF-ACSM nitrate signal into NO3,inorg and NO3,org) making the reading complicated. Overall, the results presented in the manuscript are worth publishing in ACP after clarification of several critical issues.

Major comments: - Although the main instrument of the manuscript is a ToF-ACSM and an aethalometer, a detailed description of the black carbon and its related sources is made in each section but the authors never seriously discuss the organics. I can imagine that the authors are preparing a paper dedicated to organic source apportionment, but it is a shame to present the different BC sources without mentioning those of organics. At least the authors can use the time series of well-known tracers (e.g. m/z 57 for HOA, m/z 60 for BBOA, m/z43, and 44 for the OOA, as well as m/z 79 for MSA?)

as well as look for a possible cooking aerosol contribution using the triangle approach from Mohr et al. (2012). This will certainly facilitate the interpretation of the results. - One of the most interesting and important points of this paper is certainly the contribution of ship emissions to the sulfate budget. This is the specificity of the sampling place which combines urban and ship emissions at the same place. Did the authors also consider the possible influence of ship emissions on the nitrate budget? It is known that ships are also an important source of NOx too. - How does the sea/land-breeze cycle affect the aerosol particle chemical composition and their diurnal profiles? At least during the summer months, the change in wind direction seems to have a pronounced diurnal variation ranging from 250 to 50 degrees (figure S11), which should correspond to the sea/land-breeze cycle. - section 2.1 Marseille Supersite: The discussion of the air quality parameters (O3, NO2, SO2, PM10, PM2.5, and BC) over the last 11 years is already a result in itself, which would be preferable to include in a dedicated section (e.g. overview of the general air quality, or trend on the air quality). Moreover, I would have preferred here more details regarding the sampling method itself in addition to the instrumental description. For example, were all the online instruments (ToF-ACSM, aethalometer, SMPS, BAM, FIDAS) connected to the same sampling line? Which type of inlet was used (PM)? What was the high of the inlet? How was the relative humidity controlled for each instrument (not only for the ToF-ACSM)? If the instruments were connected to the same inlet, how was the main flow distributed between them? Such information is mandatory when presenting a new sampling site. Additionally, how were the filters conditioned before and after being sampled? Which instruments were used for OC/EC and water-soluble ions measurements (manufacture, column, eluant, ...). - The authors should pay more attention to the homogeneity of the methodology applied to the manuscript. For example, regression fits are performed using orthogonal distance on the main text, which is appropriate for considering uncertainties on both datasets, but least squares regression is applied in the supplementary information. I also don't think all regression parameters (slope, intercept, and $r^2$) need 4 decimals digits. Two should be more than sufficient here. Finally, it would be nice to specify at

least once on the manuscript, "a" and "b" or to replace them with "slope" and "intercept" The same kind of comments can be made regarding wind analysis, where NWR and SWIN-2 are used. A single method will make the comparison between the results much easier and robust.

- Section 3.2.1: as mentioned above, this section is very confused and the discussion on the seasonal variability of the aerosol chemical composition is mixed with other results such as the average chemical composition over the 14-months of ToF-ACSM measurements, the average of the decade PM2.5 filter measurements, and the discussion on the aethalometer BC source estimation with no direct link to seasonal effect. Same comment for the field campaign carried out on Kaddouz site. Does it make sense to present these results here or in this paper at all? In case the authors want to keep it inside their manuscript, the sampling site must be appropriately presented in section 2, including the exact measurement period, the description of the sampling line, the list of the instruments deployed. The results must be also presented in a dedicated section and referred to in the abstract and the conclusion. Moreover, the discussion on the seasonal variation of the different aerosol compounds should be reorganized to make it consistent. For example, BC sources are discussed using the aethalometer measurements and at the end again using the UFP number concentration. The discussion will be strongly improved by combining these two parts. What about the organics? They are poorly discussed, whereas sulfate and chloride are not discussed at all. - line 438: based on which criteria the two selected case studies were selected from the 15 exceedance days? More detail will be helpful to better describe how similar were these events and discussed the factors promoting the exceedance days. Also, 2 exceeding events were selected, air mass trajectory analysis on a third one is also included. Is there any reason for that? How are the air mass trajectories for the first event? Because the 2 selected exceedance events have similar wind direction, ambient temperature, and planetary boundary layer level, it would be helpful to also discuss the wind speed during each of them or looking at the CWT profile as presented for the sulfate cases, to better understand why the first one may be considered as under

the influence of local emissions and not the second one. As the authors mentioned, the ratio BC/SO4 has some limitations which could be easily reached at the sampling according to local SO2 emissions as well as long-range transport of BC. How did the author deal with these limitations to conclude that 40 % of the exceedance days account for the local origin and 60 % for long-range transport? More explanations are strongly required. - section 3.3.2: the cluster analysis is relatively surprising here. First of all, the definition of the sea-breeze cluster is not fitting with the south-western wind sector defined line 10, and the trajectory density in Figure 11, corresponds to a land-breeze rather than a sea-breeze. Then, the discussion is focused on local processes, therefore wind direction may be more efficient for distinguishing the different wind regimes associated with such processes. What about the sea/land-breeze cycle effect as can be seen in figure S11? Is a frequency of 19 % (cluster 3) negligible? It would rather be important to compare aerosol properties (chemical composition and size distribution) during mistral and sea-breeze clusters since both are coming from the same area. -section 3.1 and Fig. 2 & S3: Is there any seasonal effect on the comparison between ToF-ACSM-BC and off-line/FIDAS/SMPS measurements? Some deviations can be seen in the comparison with SMPS and FIDAS.

Minor comments: - All acronyms must be defined before being used for the first time, even on the abstract (for example, OA and BC (line 20), UFPs (line 28), EU (line 39), LCE (line 90)). - line 13: Could you please mention the country? - line 42: 300000. - line 48: Do the authors speak about PM2.5 or PM1? - line 51: Pandolfi et al. (2020) - line 102: The dominant wind directions mentioned in the text are not visible in Figure S1. Moreover, the sea breeze wind direction is defined on the 190-270°, while in section 3.3.2, the authors named an air mass cluster "sea-breeze" having almost a pure continental origin. This is confusing. - line 134: replace lpm by L min-1 - line 138: correct Igro -line 139: Wavemetrics - line 144: How was selected the CE = 0.47? - line 167 and 170: Please check the date notation over the manuscript (with or without a 0) - line 189: How accurate is the HYSPLIT model at such a low altitude (64 m above ground level)? - line 200: How many filters were used (45 or 46)? - line 200:

Correlation coefficients are written $R^2$ on the main text and $r^2$ inside the figures. Please correct accordingly. - Line 203 acidity plot: Is there a possible seasonality effect? What happens during periods with strong deviation? For example, at the beginning of the campaign (green period) when NH4_meas strongly deviate from NH4_pred? Is there any sea-salt detected? Furthermore, it would be great to mention the different urban sources of ammonia like diesel cars. How the correlation is improved when using NO3inorg? - line 210: Please include a reference to Figure S3 when discussing the OC vs. organics. It would be also extremely interesting to compare the OC from the filter with the OC estimated from the ToF-ACSM based on the f44 signal as it can be done for the AMS (Canagaratna et al., 2015). - line 227: Please indicate the value of the selected organic density finally chosen. - line 240: Why forcing the intercept to zero here? - line 241: This conclusion can also be supported by comparing PM1 and PM2.5 from the FIDAS for the last months of the campaigns. Is this ratio constant over the 14-months? Is there any seasonal or diurnal variation on the ratio? - line 247: Is the PM1 refers here to the ACSM-BC? - line 248: Would it be simpler to always refer to the same recommendation of the WHO? Here it is 10 $\mu$g m-3, in Figure 3 it is 25 $\mu$g m$^{-3}$, as well as for the selection of the case studies. - line 253: Which factor was used for the conversion of the OC to OM? - line 312: Can refer to Schaap et al. (2004) for example. - line 321: Can the summer results be influenced by the low nitrate mass concentration at this time of the year? Which lowest detection limit was used here? Is there any link between the NO3,org and BCwb as the aging of wood-burning aerosol can lead to nitrogen-containing compounds? - line 384: Is it still related to BCwb? - line 393: Is there any reason why summer NO3,org is only discussed for June 2017? - line 413: How does it compare with the organics or m/z44? - line 419 – 420: Could you please detail a bit more? What does it mean "the N2(10-20 nm) number concentration, corresponding to 90% of the total number in this range"? - line 426: Does the PM1 mass concentration of the selective days also exceed the 25 $\mu$g m-3 over 24h? - line 431: Is there any explanation for the 2 outstanding years? Could it be related to specific weather conditions or local events? - line 435: This is

quite difficult to see in Figure 1. - line 455: Particle number size distribution during the selected event would be helpful. - line 470: "the aerosol chemical composition was relatively stable" - line 497: Could it be possible that the polluted air masses were rather coming from the Rhone valley than bringing Pô valley polluted air masses over the Alpes mountains? Can a trajectory analysis (CWT) help to identify the potential aerosol source area? - line 498: Why is there a new case study event? - line 519: Is there any confusion here? This section aims to discuss the summer sulfate origin and figure 8 the relation between sulfate concentration and UFP over the seasons. This should rather be done earlier on the seasonal analysis part. - line 525: Please rephrase the sentence "during summertime, the ships traffic increases by 25 % (4319 against 3263 for the 2017-2017 period)". What are the numbers referring to? - line 530: I disagree a bit here since the SO2 concentration is continuously increasing from midnight to 9 o'clock, so much earlier than the ship traffic peak. - line 556: Is the discussion on the sulfate classification needed? It was already mentioned that sulfate is fully neutralized by ammonium. No new conclusion was drawn from this sulfate fragmetns analysis. It would be interesting here to look for example, at the time series of the 3 different sulfate species and compare the MSA results with the time series of the m/z 79. - line 559: Numbering of figures 11 and 12 should be changed. Figure 12 is discussed first. - line 574: The term long-range transport is relative here since sulfate sources look to the located relatively close to the city of Marseille.

Tables, Figures, and supplementary information: - Please used scientific notation on the axis labeling. -Figure 2-a: Did the authors investigate the deviation between ACSM-BC and SMPS when density increase? Seems that there is a deviation for density above 1.5. Could it be linked to the presence of more sea-salt or coarse particles? - Figure 6: Is there any reason why BCwb and BCff are presented in Fig 6a and not in Fig 6b? Wind speed would also be interesting here. - Figure 8: What does the number of points mean (time resolution)? Moreover, a log-scale would be helpful to better catch the number of points on each category. - Figure 9: The difference between the two red colors (arrival and departure at the South terminal) is not easy to catch. Please, provide

information for the wind speed intensity value. - Figure 10: Please include similar plots for the missing cluster 3. - Figure 12: Please include the expected limits of the triangle (location of organosulfates, MSA and ammonium sulfate). Ammonium sulfate from the 7/12 calibration is missing. -Figure S1: The central map is rather too small and it is very difficult to distinguish the different colors on the Marseille port. - Figure S3 caption: Please, correct PM1 notation. - Figure S5: please include the zero lines - It will make the reading of the supplementary information easier by including tables and figures directly in the corresponding text section. - Is figure S14c discussed?

References: -Canagaratna, M. R., Jimenez, J. L., Kroll, J. H., Chen, Q., Kessler, S. H., Massoli, P., Ruiz, L. H., Fortner, E., Williams, L. R., Wilson, K. R., Surratt, J. D., Donahue, N. M., Jayne, J. T., and Worsnop, D. R.: Elemental ratio measurements of organic compounds using aerosol mass spectrometry: characterization, improved calibration, and implications, Atmos. Chem. Phys., 15, 253-272, doi 10.5194/acp-15-253-2015, 2015. -Mohr, C., DeCarlo, P. F., Heringa, M. F., Chirico, R., Slowik, J. G., Richter, R., Reche, C., Alastuey, A., Querol, X., Seco, R., Penuelas, J., Jimenez, J. L., Crippa, M., Zimmermann, R., Baltensperger, U., and Prevot, A. S. H.: Identification and quantification of organic aerosol from cooking and other sources in Barcelona using aerosol mass spectrometer data, Atmos. Chem. Phys., 12, 1649-1665, 2012. -Schaap, M., Spindler, G., Schulz, M., Acker, K., Maenhaut, W., Berner, A., Wieprecht, W., Streit, N., Muller, K., Bruggemann, E., Chi, X., Putaud, J. P., Hitzenberger, R., Puxbaum, H., Baltensperger, U., and ten Brink, H.: Artefacts in the sampling of nitrate studied in the "INTERCOMP" campaigns of EUROTRAC-AEROSOL, Atmos. Environ., 38, 6487-6496, 10.1016/j.atmosenv.2004.08.026, 2004.

---

## Referee Comment (RC2) · Anonymous Referee #2 · 15 Dec 2020

Review of the measurement report manuscript by Chazeau et al. This manuscript focus mainly on aerosol measurements conducted from Feb-2017 till April-2018 at a downtown site in Marseille. The aerosol data site combined ACSM, Aethalometer, size distribution, and ions from 24-h filters. Regulated gases where also measured at the site.

The manuscript sets-out to characterize PM1, atmospheric dynamics and a few pol-

lution events captured during this period, however it cannot be said to achieve its objective completely. I cannot recommend publication of the manuscript as it stands and provide a detailed list of aspects that need to be carefully improved.

General comments:

1) The manuscript is quite lengthy to read, and at the same feels missing the target due to over simplistic analysis. Furthermore, claims missing references and/or lacking precision are found through the text. I would recommend the entire text to be carefully revised by the authors to improve its general quality. I have added several points as technical comments, but please do not be restrict yourselves to what has been pointed out, as the manuscript would benefit from a careful review from the experienced authors.

2) I do not agree with the authors' use of "long-term" on the title and throughout the text. Nowadays some Europeans sites are pushing 10 years of comparable instrumentation (ACSM, Aeth, SMPS), so in the scientific context, one year does not match the definition. In the text, even size distribution is mentioned to be long-term here, which is of course out of touch with the community. I'd recommend to remove every single mention of it in the text for a better description of the dataset.

3) I find the lack of PMF analysis of ACSM becomes a handicap for the interest of the paper. The manuscript itself brings forth questions of $NO_3$,org or BCff/BCwb which can be answered (or at least hypothesized) with an statistical analysis of organic spectra, but are not presented here. Further problems rise from specific pollution events that could be enriched by that analysis. As it stands I find the manuscript somewhat frustrating, and would significantly benefit by going this extra mile.

4) Finally, I'm not particularly fond of the title, after adding the "measurement report:" now you have two sets of colons on it. I'd suggest to remove the "tracking the polluted..." part of it.

Specific comments:

5) The UFP analysis on N1/N2 is not very thorough and can suffer from large bias (change in meteorological conditions, specific sources) which are not necessarily captured by linear correlation with BCff. If the interest of the authors is to exploit the freshly nucleated particles, I'd suggest to focus only on the N10-20 fraction as the end of 3.2 section, even if the statistics are smaller.

6) Section 3.3.1: This analysis of two pollution events do not add significantly to the publication, or generally to understanding pollution events to the site. Typically those are fairly well represented by operational atmospheric models, so the question would be how close known processes and inventories represent them, rather than relying only on in-situ aerosol measurements and local winds for this type of analysis.

7) Section 3.3.2: This section reads like a patch of several analysis hardly including more than a paragraph and a figure. The clearer example is Figure 12 and SO4 ion analysis. What is the goal with this analysis, in the context of this section and this manuscript more generally? The next two points also relate to this section.

8) Cluster analyses: I'm reticent about the use of hysplit cluster on interpreting such short-scale air masses movements, particularly sea breezes. I agree that on average, the continental cluster might be more prone to sea breeze and thus be continuously fed by anthropogenic emissions, but it's a big step naming it "sea breeze", as the same can happen with Mediterranean air masses. I don't see so clearly the "discernible" differences on figure 10, which can be just due to the small statistics treated here.

9) On the cluster topic, given possible role of local sources, analysis such as figure 11 might be extremely misleading. From a quick look it seems that CWT maps follow trajectory densities. My guess would be that you'd fine similar maps for locally emitted BCff maps, for example.

Technical Check US vs UK English spelling, both are found in the text. L.46: replace

"leave" by "live". Reference for this claim? L.51: please replace "(Pandolfi et al., 2020)" by "Pandolfi et al., (2020)" L. 72: please remove comma between "particles" and "have". L.90: Please define LCE L.95-96: I don't find this sentence overly clear. Does it mean it is a busy downtown area? Of course comparing with the national average this type of environment should drive the average up. Please make it clearer, and add the reference for such claim. L.97: replace "first" by "largest" L.98: "berthing almost 4000 ships in 2017." Also without reference. L.102: "Driven" instead of "held"? L.108-111: Missing reference for this sentence. L.133: "The graphs on the right side display" instead of "The right graphs display". L.123: please replace "very slight" by a percentage. L.159: "ACSM" instead of "ACMS". L.241: "confidence" interval. L.259-L261: I'd suggest to remove this sentence with the reference to previous work, as at this point of the manuscript there is only general description of chemical composition and no insights into their origins. L.260: C-ToF-AMS has not been defined. Fig.3: The pie charts on Fig. 3 seem to suffer from low resolution. In addition, the font size indicating the seasons should be increased. The date stamp should be in the format dd-mmm-yy to avoid confusion. L.264-266: Careful with the comparison, the fractional contribution of PM1-NR + BC and filter based PM2.5 are not directly comparable. The results from Putaud were from filter (wider range of species detected, especially in the refractory range such as SS and dust), and from some 15 years ago. L.266-267: Unclear why this sentence (and ref) has been added here. L.267: "average" L.292: I fail to see a clear seasonality on OA concentrations, from both the plot and Table 1, so perhaps re-write this sentence. L.302: replace "big" by "large". L.303: remove "still". L.343-344: I find this sentence to be imprecise. It's missing reference for the link between UFP and health, and UFP can grow under the right conditions to significantly impact PM1, PM2.5 or PM10. I'd suggest to just remove it. L.350-351: Was this seasonal variation linked to BC and OA to be observed from Fig. 3? No indication whatsoever can be seen from it. L.429-431: missing reference. L.435-436: rewrite sentence. L.437-438: Unclear why Christmas event would be local pollution driver. Fig. 6: dates for Christmas event not the same as the text. Fig. 6: change time stamp for the format dd-mmm-yy. L.514-515:

repetition of information. Figure S11: Correct caption N(10-20nm)

---

## Author Comment (AC1) · 1 Mar 2021

**General response to reviewers:**

We thank both anonymous referees #1 and #2 for providing constructive and helpful reviews. We have implemented all these suggestions and comments in the revised version of the manuscript. The line-by-line responses to the reviewers' comments (written in black) are written in blue. We hope that these improvements meet the reviewers' expectations.

As the general remark both referees have encouraged us to carefully revise the manuscript in order to emphasize the scientific questions we aim to address in this study.

The main goals of this study were to better characterize the phenomenology of $PM_1$ in Marseille, representative of a coastal city in western Mediterranean, and attempt to identify its specific features. To achieve these goals, we have used the continuous measurements of $PM_1$ composition provided by the ToF-ACSM over the first 14-month following its implementation in Marseille, in association with several additional key pollutants ($NO_x$, $O_3$, $SO_2$, $PM_{10}$, $PM_{2.5}$, BC, Ultrafine particles) installed in the new supersite MRS-LCP. Most important findings are that 1) we could experimentally determine whether and to which extent local and/or remote pollution contribute to the $PM_{2.5}$ daily mean threshold (25 µg m$^{-3}$) exceedance observed at the site 2) we have demonstrated the contrasting particle chemical composition between these local/remote sources of pollution, and 3) we could identify a strong local source of sulfate related to industrial and shipping activities in the nearby coastal area in summertime, in addition to a regional source from the Mediterranean basin strongly believed to originate from shipping. We have thoroughly revised the manuscript in order to highlight these findings.

**REFEREE #1**

Review Chazeau et al. describe a 14-months measurement campaign at the urban station of Marseille-Longchamp supersite (Marseille, France) from February 2017 to April 2018. The paper focuses on the analysis of data from ToF-ACSM and aethalometer measurements. Besides, it is important to note that the station is also used by the local air quality agency providing long-term measurements of standard air quality variables (NOx, O3, SO2; BC, PM10, and PM2.5), which are partly included in the manuscript. The discussion begins with a mass closure analysis using collocated instrumentations, following by a description of the seasonal and the diurnal profiles of the PM2.5 chemical components. Two case-study events corresponding to periods when the total PM mass exceeded the WHO recommendation of 25 µg/m3 over 24 h were considered and described in more detail (Christmas 2017 and 4 consecutive days in February 2018). Finally, the results section ends with a discussion on the influence of ship and industry emissions on the local sulfate mass concentration. Although the manuscript responds to the need for a +1 year continuous measurements at a high time-resolution to better characterize the different factors and sources influencing local air quality, the results are presented rather as a descriptive report than as an attempt to answer a well-identified scientific question or to focus on a specific topic. This lack of a central theme makes it difficult to read and to catch the link between the different sections and the sub-sections. A direct consequence is that sometimes, I had the impression that the authors have lost their focus and started to describe results that are not directly related to the measurements made at the Marseille-longchamp supersite (for example, the discussion about BC source at the Kaddouz site) or to change the subject before returning to it (for example, the seasonal variability section starts and ends on BC source estimation, with in-between results from the splitting of the ToF-ACSM nitrate signal into NO3,inorg and NO3,org) making the reading complicated. Overall, the results presented in the manuscript are worth publishing in ACP after clarification of several critical issues.

**Major comments:**

- Although the main instrument of the manuscript is a ToF-ACSM and an aethalometer, a detailed description of the black carbon and its related sources is made in each section but the authors never seriously discuss the organics. I can imagine that the authors are preparing a paper dedicated to organic source apportionment, but it is a shame to present the different BC sources without mentioning those of organics. At least the authors can use the time series of well-known tracers (e.g. m/z 57 for HOA, m/z 60 for BBOA, m/z43, and 44 for the OOA, as well as m/z 79 for MSA?) as well as look for a possible cooking aerosol contribution using the triangle approach from Mohr et al. (2012). This will certainly facilitate the interpretation of the results.

We would like to thank the reviewer for these suggestions. We are indeed preparing a paper specifically dedicated to the source apportionment of organic aerosol with a new PMF (Positive Matrix Factorization) approach. Following the reviewer's recommendation, the mentioned specific organic fragments are now included in the revised manuscript: $f_{44}$, $f_{55}$ $f_{57}$ and $f_{60}$ are used to assess the sources of the seasonal and daily variability of the $PM_1$ in section 3.3.2 and 3.3.3 and $f_{44}$ and $f_{60}$ are used to discriminate between the local and the regional/long range pollution episodes in section 3.4.1.

- One of the most interesting and important points of this paper is certainly the contribution of ship emissions to the sulfate budget. This is the specificity of the sampling place which combines urban and ship emissions at the same place. Did the authors also consider the possible influence of ship emissions on the nitrate budget? It is known that ships are also an important source of NOx too.

Some intense $NO_x$ peaks, concomitant with $SO_2$ have been occasionally observed under a sea breeze system, as shown in Figure A1a, and could possibly be attributed to ship traffic emissions. However, the ship contribution to the $NO_x$ concentration is negligible compared to the traffic source, as demonstrated by the bimodal pattern $NO_x$ diurnal cycle, typical of road traffic in Figure A1b. During these events, a slight increase of the particulate nitrate concentration can only be observed. These observations suggest that the ship impact on the nitrate budget in the city can be considered negligible.

[Figure]

**Figure A1 – Time series of wind speed, wind direction, $SO_2$, $NO_x$, $SO_4^{2-}$ and $NO_3^-$ concentrations from 07 to 09 July 2017. (a). Mean diurnal cycle of $NO_x$ during summer 2017 (b).**

- How does the sea/land-breeze cycle affect the aerosol particle chemical composition and their diurnal profiles? At least during the summer months, the change in wind direction seems to have a pronounced diurnal variation ranging from 250 to 50 degrees (figure S11), which should correspond to the sea/land-breeze cycle.

The land-sea breeze cycle impacts the air quality in Marseille in two different ways. In summer, the sea breeze blows inland the coastal emission sources during the day and gives rise essentially to an increased sulfate concentration (see section 3.4.2) and a higher frequency of occurrence of short term high UFPs concentration episodes (see section 3.3.2 and 3.4.2), in connection with the $SO_2$ emission from industrial and shipping activities. In winter, the land breeze that sets in the evening blows to the city the biomass burning emissions that come from the surrounding areas (located in the North and north-East of the city) (see sections 3.3.2 and 3.4.1). In that case the organic matter can make up most of the $PM_1$ mass (80-100%) and the 25 µg m$^{-3}$ 24-hour mean from the WHO guideline for particulate matter is exceeded.

- section 2.1 Marseille Supersite: The discussion of the air quality parameters (O3, NO2, SO2, PM10, PM2.5, and BC) over the last 11 years is already a result in itself, which would be preferable to include in a dedicated section (e.g. overview of the general air quality, or trend on the air quality). Moreover, I would have preferred here more details regarding the sampling method itself in addition to the instrumental description. For example, were all the online instruments (ToF-ACSM, aethalometer, SMPS, BAM, FIDAS) connected to the same sampling line? Which type of inlet was used (PM)? What was the high of the inlet? How was the relative humidity controlled for each instrument (not only for the ToF-ACSM)? If the instruments were connected to the same inlet, how was the main flow distributed between them? Such information is mandatory when presenting a new sampling site. Additionally, how were the filters conditioned before and after being sampled? Which instruments were used for OC/EC and water-soluble ions measurements (manufacture, column, eluant, ...).

Following the referee recommendation, the section describing the air quality parameters was removed from section 2.1. It is now discussed in a new dedicated section (3.1: "Air quality overview") in "3 Results and discussion".

Section 2.2.2 now includes additional information on the sampling systems: inlet configuration, sampling flow rate and conditioning are reported. Filters conditioning and analysis protocols are also detailed.

- The authors should pay more attention to the homogeneity of the methodology applied to the manuscript. For example, regression fits are performed using orthogonal distance on the main text, which is appropriate for considering uncertainties on both datasets, but least squares regression is applied in the supplementary information. I also don't think all regression parameters (slope, intercept, and r2 ) need 4 decimals digits. Two should be more than sufficient here. Finally, it would be nice to specify at least once on the manuscript, "a" and "b" or to replace them with "slope" and "intercept" The same kind of comments can be made regarding wind analysis, where NWR and SWIN-2 are used. A single method will make the comparison between the results much easier and robust.

We agree that orthogonal distance should be used for all regressions in the manuscript and we have modified the revised version accordingly. The number of decimal digits was reduced to 2 and "a" and "b" terms were specified in the revised manuscript as follows: *"Orthogonal distance regressions were performed for the analyses and the term "a" and "b" referred to the intercept and the slope, respectively."*

Regarding the wind analysis, the NWR method was used except for the pollution episode that occurred in February 2018, where the sustained wind incidence method (SWIM) was used instead. As mentioned in the text, the North/North-West wind sector was clearly dominant during this event and

NWR analysis was biased because of high standard deviation induced by sporadic changes in wind direction (Figure A2). In such a special case, we believe the SWIM-2 analysis was a better alternative to NWR. This is specified in the manuscript (section 2.3).

[Figure]

**Figure A2 – NWR (top) and SWIM-2 (bottom) analyses for NO$_{3,Inorg}$/BC ratio and f44 contribution during the pollution episode of February 2018.**

- Section 3.2.1: as mentioned above, this section is very confused and the discussion on the seasonal variability of the aerosol chemical composition is mixed with other results such as the average chemical composition over the 14-months of ToF-ACSM measurements, the average of the decade PM2.5 filter measurements, and the discussion on the aethalometer BC source estimation with no direct link to seasonal effect. Same comment for the field campaign carried out on Kaddouz site. Does it make sense to present these results here or in this paper at all? In case the authors want to keep it inside their manuscript, the sampling site must be appropriately presented in section 2, including the exact measurement period, the description of the sampling line, the list of the instruments deployed. The results must be also presented in a dedicated section and referred to in the abstract and the conclusion. Moreover, the discussion on the seasonal variation of the different aerosol compounds should be reorganized to make it consistent. For example, BC sources are discussed using the aethalometer measurements and at the end again using the UFP number concentration. The discussion will be strongly improved by combining these two parts. What about the organics? They are poorly discussed, whereas sulfate and chloride are not discussed at all.

Following the reviewer remarks, the whole section 3.2 has been substantially rewritten and reorganized. In the revised manuscript, Section "3.3 Submicron aerosol temporal variability" is split into three sub-sections: "3.3.1 PM Average composition", "3.3.2 Seasonal variability" and "3.3.3 Diurnal profiles". The discussion has been extended on organics (using f$_{44}$, f$_{55}$, f$_{57}$, and f$_{60}$ markers), and sulfate, ammonia and chlorides are now included in the discussion.

Sub-section 3.3.2 now only focuses on seasonal trends, meaning that the approach to determine the angstrom exponent using data from the urban kerbsite (Kaddouz) was moved to the Supplement. Meanwhile, in this section, we have chosen not to mix the discussion between the chemical composition and the UFPs as suggested by the referee. Instead, we have modified the section dedicated to the UFPs so that the term "BC" only appears when needed, which is in the equation used to determine $N_1$ and $N_2$.

- line 438: based on which criteria the two selected case studies were selected from the 15 exceedance days? More detail will be helpful to better describe how similar were these events and discussed the factors promoting the exceedance days. Also, 2 exceeding events were selected, air mass trajectory analysis on a third one is also included. Is there any reason for that? How are the air mass trajectories for the first event? Because the 2 selected exceedance events have similar wind direction, ambient temperature, and planetary boundary layer level, it would be helpful to also discuss the wind speed during each of them or looking at the CWT profile as presented for the sulfate cases, to better understand why the first one may be considered as under the influence of local emissions and not the second one. As the authors mentioned, the ratio BC/SO4 has some limitations which could be easily reached at the sampling according to local SO2 emissions as well as long-range transport of BC. How did the author deal with these limitations to conclude that 40 % of the exceedance days account for the local origin and 60 % for long-range transport? More explanations are strongly required.

Even if $BC/SO_4^{2-}$ ratio is a proxy that can be used to discriminate between local and remote pollution (Petit et al., 2015), it has some limitations. The revised manuscript now includes a comprehensive list of criteria used to distinguish the local from the long-range transport during the days exceeding the $PM_{2.5}$ WHO recommendation. The list includes: $f_{44}/f_{60}$ ratio (also inspected in Figure 9), BC contribution to $PM_{2.5}$, nitrate contribution to $PM_1$, wind speed and $\Delta PM_{2.5 \text{ (Land breeze/Other winds)}}$ (Table S5). The probability density distributions of these parameters during the $PM_{2.5}$ exceedance days are shown in Figure 10.

Table S5 was added to the Supplement and Figure 9 and Figure 10 were added to the revised version of the manuscript.

- section 3.3.2: the cluster analysis is relatively surprising here. First of all, the definition of the sea-breeze cluster is not fitting with the south-western wind sector defined line 10, and the trajectory density in Figure 11, corresponds to a land-breeze rather than a sea-breeze. Then, the discussion is focused on local processes, therefore wind direction may be more efficient for distinguishing the different wind regimes associated with such processes. What about the sea/land-breeze cycle effect as can be seen in figure S11? Is a frequency of 19 % (cluster 3) negligible? It would rather be important to compare aerosol properties (chemical composition and size distribution) during mistral and sea-breeze clusters since both are coming from the same area.

We agree that there was a potential for confusion on the so-called "sea breeze" cluster. This cluster was actually related to situations where initial continental air masses were blown along the Rhone Valley seaward, and returned inland when the sea breeze sets in. Figure A3 demonstrates the progressive anticlockwise rotation that occurs on August 7, 2017, during the day as the sea breeze regime is accentuated. A good agreement was found with the wind measurements at the station. For this cluster, the CWT analysis on $SO_4^{2-}$ clearly showed a hotspot over the sea breeze area.

[Figure]

**Figure A3 – Hourly backtrajectories arriving in MRS-LCP on august 7, 2017.**

It is recognised, however, that the backtrajectory model is not expected to capture the small scale structure of sea breeze system, as already highlighted by Drobinski et al. (2007). In the present work, we fully understand the point raised by the referee since such local wind regimes as sea breeze were found in both clusters.

As a consequence, we decided to change our clustering method by a k-means clustering analysis of sulfate hourly concentrations for each summer day. From this sulfate classification, we investigated the different wind regimes and related backtrajectories. Results of this analysis are presented in section 3.4.2. Further details are also provided in the responses to the anonymous referee #2.

-section 3.1 and Fig. 2 & S3: Is there any seasonal effect on the comparison between ToF-ACSM-BC and off-line/FIDAS/SMPS measurements? Some deviations can be seen in the comparison with SMPS and FIDAS.

Since SMPS and FIDAS never operated over the same period, it makes the inspection of the slight slopes deviation (slope of 1.02 and 0.9, when the ACSM+BC is fitted with the SMPS and FIDAS respectively) rather speculative. However, we further investigated the possible seasonality in $PM_1/PM_{2.5}$ ratio and the short deviation period on the ACSM+BC vs. SMPS correlation (see the responses to the related reviewer's comments at page 10 and 16, respectively).

**Minor comments:**

- All acronyms must be defined before being used for the first time, even on the abstract (for example, OA and BC (line 20), UFPs (line 28), EU (line 39), LCE (line 90)).

Acronyms for OA, BC, UFPs, EU and LCE are defined in the revised manuscript.

- line 13: Could you please mention the country?

The country is mentioned in the revised version.

- line 42: 300000.

Corrected.

- line 48: Do the authors speak about PM2.5 or PM1?

Both $PM_{2.5}$ and $PM_1$ are discussed. Although European standards exist for $PM_{2.5}$, they have not been transposed into the French legislation yet. But very recently (1$^{st}$ January 2021), a new agreement integrating the $PM_{2.5}$ measurements to the calculation of the French air quality index has been adopted (https://www.ecologie.gouv.fr/nouvel-indice-atmo-plus-precis-et-plus-clair). The sentence in the revised version is now only focused on the $PM_1$.

- line 51: Pandolfi et al. (2020)

The text was replaced.

- line 102: The dominant wind directions mentioned in the text are not visible in Figure S1. Moreover, the sea breeze wind direction is defined on the 190-270° , while in section 3.3.2, the authors named an air mass cluster "sea-breeze" having almost a pure continental origin. This is confusing.

To improve the representation of the wind characteristics in Marseille, the joint probability plot has been replaced by the rose plot (Figure A4) in the revised manuscript. The text has been modified as follows:

*"Moreover, Marseille air quality is often affected by two regional winds (Mistral and South-Easterly Mediterranean wind) and local sea/land breeze cycles. The Mistral is a strong wind blowing from the North-West (300°-360°) along the lower Rhône River valley toward the Mediterranean Sea. South-Easterly Mediterranean wind (105°-135°) blows at similar intensity from the sea toward the lands. The South-Westerly sea breeze (190°-270°) (210°-270°) and North-Easterly land breeze (5°-90°) are local winds prevailing during weak Mistral wind (Figure S1).*

[Figure]

**Figure A4 - Rose plot of wind speed and wind direction is represented for the full study period.**

- line 134: replace lpm by L min-1

The term was corrected.

- line 138: correct Igro

The term was corrected.

-line 139: Wavemetrics

The term was corrected.

- line 144: How was selected the CE = 0.47?

CE = 0.47 is only given for comparison purpose. It is the average value over the entire period of the time dependant CE. The sentence *"For this dataset CE is assessed as 0.47±0.05 which is comparable to values typically found for ambient aerosol (0.5, Middlebrook et al., 2012)."* was replaced by *"On average over the entire period, a composition dependant CE of 0.47 ±0.05 is obtained, which is comparable to values typically found for ambient aerosol (0.5, Middlebrook et al., 2012)."*

To make it clear, it is the time and composition dependent collection efficiency (CDCE) shown in Figure S2 that was applied to the dataset.

- line 167 and 170: Please check the date notation over the manuscript (with or without a 0)

Thank you for mentioning this. We have harmonized the date and time notations throughout the whole manuscript.

- line 189: How accurate is the HYSPLIT model at such a low altitude (64 m above ground level)?

We agree with the reviewer that results from HYSPLIT model at such a low arrival height should be interpreted with caution. To increase the confidence of our analysis, back trajectories model was run for three different arrival altitudes: 64m, 100m and 500m AGL. As shown in Figure A5, the trajectories are not very sensitive to the height of arrival. Since MRS-LCP site is located at 64 m AGL, this value was chosen as arrival altitude in the HYSPLIT model.

[Figure]

**Figure A5 - HYSPLIT air mass 72h-backtrajectories during February 2018 polluted event at three different arrival levels: 64m (left), 100m (middle) and 500m (right) AGL.**

- line 200: How many filters were used (45 or 46)?

The number of filters was changed to 45, thanks.

- line 200: Correlation coefficients are written R2 on the main text and r2 inside the figures. Please correct accordingly.

All correlation coefficients are now written "$R^2$" in the main text and figures.

- Line 203 acidity plot: Is there a possible seasonality effect? What happens during periods with strong deviation? For example, at the beginning of the campaign (green period) when NH4_meas strongly deviate from NH4_pred? Is there any sea-salt detected? Furthermore, it would be great to mention the different urban sources of ammonia like diesel cars. How the correlation is improved when using NO3inorg?

We further inspected the short period noticed by the reviewer (03/02/2018, 18h00 to 04/02/2018, 17h00; green period in the Figure S4) but it is not clear why the measured ammonium is greater that the predicted. This would imply that at this specific period, $NH_4$ was associated to species other than the one accounted for in the default balance calculation (sum of $NH_4Cl$, $NH_4NO_3$ and $NH_4(SO_4)_2$). It is noteworthy to note that at that time, sulfate and chloride concentrations were close to their detection limits, as was the case for $NO_2^+$ (Figure A6). In the meantime, the $NO^+$ fragment equivalent concentration remained significant (average of 0.73 µg m$^{-3}$). While this situation could be indicative of the presence of sea salts ($NaNO_3$, $Mg(NO_3)_2$ $Ca(NO_3)_2$ generate very low $NO_2^+/NO^+$ ratio (Farmer et al., 2010), this hypothesis does not support the deviation of $NH_{4meas}$. These salts would induce an overestimation of the $NH_{4pred}$ (calculated from the signal produced by the $NO_3$ fragments) while not being counterbalanced by some signal produced at $NH_4^+$, resulting in a lower $NH_{4meas}/NH_{4pred}$. The same argument applies for the organic nitrates as they also produce low $NO_2^+/NO^+$ ratio. To conclude based on our current analysis, we cannot state on the origin of the deviation.

[Figure]

**Figure A6 - Time series of NH4$_{meas}$/NH4$_{pred}$ ratio, NO$_2^+$ and NO$^+$ fragments from ToF-ACSM and Cl$^-$, NH$_4^+$, NO$_3^-$ and SO$_4^{2-}$ concentrations from 3$^{rd}$ to 4$^{th}$ February 2018.**

Thanks for the suggestion on different sources of ammonia, we have added a mention on ammonia urban sources in 3.2: *"In addition to agricultural activity, ammonia can be emitted by sources closed to urban area, such as vehicular exhausts, sewage, industrial emissions or residential biomass burning (Meng et al., 2017; Sun et al., 2017; Sutton et al., 2013)."* and in 3.3.3: *"Suarez-Bertoa et al.*

*(2015) mentioned that urban traffic emissions of ammonia have increased in Europe (+378%) over the last decades leading to possible enhanced ammonium concentrations."*

We also performed the acidity plot by using $NO_{3,Inorg}$ instead of total nitrate. The slope and $R^2$ increased (1.04 and 0.97, respectively) suggesting that inorganic nitrate concentrations are mostly neutralised. Note that some data points were not included in the calculation if their signals at m/z 46 or m/z 30 data points are below detection limits. This must contribute to the improvement of $R^2$ value.

- line 210: Please include a reference to Figure S3 when discussing the OC vs. organics. It would be also extremely interesting to compare the OC from the filter with the OC estimated from the ToF-ACSM based on the f44 signal as it can be done for the AMS (Canagaratna et al., 2015).

Thanks for this suggestion. We performed this calculation and we determined an OM to OC ratio of 1.94 over the offline measurements period and 1.91 over the 14 months study period. The following text was added in section 3.2: *"A theoretical OM to OC ratio can be calculated based on fractional contribution of m/z 44 (f44) mainly due to $CO_2^+$ (Canagaratna et al., 2015). The determined ratio was 1.94 over the offline measurements period, which is lower than the ratio obtained from the OC filters comparison method, but remains elevated for an urban site. From this method, a value of 1.91 was obtained for the entire study period."*

A reference to the Figure S3 was also included in the discussion.

- line 227: Please indicate the value of the selected organic density finally chosen.

We mentioned a value of 1.2 g cm$^{-3}$ for organic density in the manuscript. For clarity, the sentence has been changed to: *"Finally, a density of 1.2 g cm$^{-3}$ was chosen for organic aerosol (Cross et al., 2007)"*.

- line 240: Why forcing the intercept to zero here?

We changed the fit for an orthogonal distance regression without forcing the intercept to 0.

- line 241: This conclusion can also be supported by comparing PM1 and PM2.5 from the FIDAS for the last months of the campaigns. Is this ratio constant over the 14-months? Is there any seasonal or diurnal variation on the ratio?

The $PM_1$ vs $PM_{2.5}$ comparison from the FIDAS measurements gave a ratio of 0.8 over its deployment period (between 19 February and 13 April 2018). A similar ratio is found for the reconstituted $PM_1$ (NR species +BC) and the PM2.5 from the BAM 1020 over the same period (=0.8).

The $PM_1/PM_{2.5}$ ratio over the 14-months slightly varied between the seasons with 0.81 for winter, 0.74 for spring, 0.85 for summer and 0.89 for autumn. High level of coarse particles can be found during spring due either to enhanced inorganic mass concentration (Bressi et al., 2013; Petit et al., 2015; Schaap et al., 2004; Squizzato et al., 2013) or higher occurrence of Saharan dust events (Querol et al., 2009). However, such events were not recorded for our dataset even if the spring ratio is lower.

The ratio did not show any diurnal pattern as a constant flat trend was obtained.

- line 247: Is the PM1 refers here to the ACSM-BC?

Yes indeed. We added this information in the revised version.

- line 248: Would it be simpler to always refer to the same recommendation of the WHO? Here it is 10 µg m-3, in Figure 3 it is 25 µg m-3 , as well as for the selection of the case studies.

Through the manuscript, we mostly refer to the WHO $PM_{2.5}$ daily recommendation. But in this particular case we compared the average $PM_{2.5}$ concentration of the study period with the annual recommendation. This was notified accordingly in the revised version.

- line 253: Which factor was used for the conversion of the OC to OM?

We used an OM to OC ratio of 1.4 given by the corresponding studies (Bozzetti et al., 2017; Salameh et al., 2015). Note that the detailed discussion about the PM chemical composition comparison with Bozzetti et al. (2017) and Salameh et al. (2015) has been removed to make the manuscript more concise and readable.

- line 312: Can refer to Schaap et al. (2004) for example.

The reference was added, thanks.

- line 321: Can the summer results be influenced by the low nitrate mass concentration at this time of the year? Which lowest detection limit was used here? Is there any link between the NO3,org and BCwb as the aging of wood-burning aerosol can lead to nitrogen-containing compounds?

We agree with the anonymous referee that higher contribution of organic nitrate in summer may be linked to the very low concentration level of total nitrate (0.24 µg m$^{-3}$ on average). We added the following sentence to the section 3.3.2: *"Still, these results might be partly influenced by the low level of total nitrate concentration encountered during this period."*

Nonetheless, $NO_{3,Org}$ and $NO_{3,Inorg}$ display distinct diurnal profiles for this season (Figure 6) and the calculation was not applied for nitrate concentrations below the detection limit (0.018 µg m$^{-3}$). The nitrate segregation is not feasible neither if m/z 46 or m/z 30 were below nor equal to 0.01 µg m$^{-3}$ making the results trustable.

As suggested by the reviewer, we inspected the possible link between $NO_{3,Org}$ and $BC_{WB}$. As shown on the Figure A7, $NO_{3,Org}$ fraction does not correlate with $BC_{WB}$.

[Figure]

**Figure A7 - $NO_{3,Org}$ vs $BC_{WB}$ over the study period. The color-scale indicates the time period.**

- line 384: Is it still related to BCwb?

Yes it is. The line break was removed for more clarity.

- line 393: Is there any reason why summer NO3,org is only discussed for June 2017?

The analysis now includes all summer data.

- line 413: How does it compare with the organics or m/z44?

The seasonal diurnal cycles of f44 are now included in Figure 6. In summer, f44 shows a similar diurnal pattern than $N_2$, temperature, ozone and sulfate. This was mentioned in the revised text.

- line 419 – 420: Could you please detail a bit more? What does it mean "the N2(10-20 nm) number concentration, corresponding to 90% of the total number in this range"?

The text was modified as following: *"In this size range, the secondary ($N_{2\ (10\text{-}20\ nm)}$) and fresh primary emissions ($N_{1\ (10\text{-}20\ nm)}$) fractions corresponded to 90% and 10% of the total number concentration ($N_{(10\text{-}20\ nm)}$), respectively."*

- line 426: Does the PM1 mass concentration of the selective days also exceed the 25 µg m-3 over 24h?

Yes indeed, it has been clarified in the revised manuscript: *"The average $PM_{2.5}$ concentrations were 31.2 µg m$^{-3}$ and 36.7 µg m$^{-3}$ respectively for the local and regional pollution episodes, with average $PM_1$ concentrations of 28.7 µg m$^{-3}$ and 31.1 µg m$^{-3}$, respectively. This indicates that $PM_{2.5}$ pollution episodes were driven by $PM_1$ concentrations in both cases."*

- line 431: Is there any explanation for the 2 outstanding years? Could it be related to specific weather conditions or local events?

A likely cause would be the higher occurrence of forest fire events during these outstanding years. The Figure A8 represents the number of fire events occurring in summer in the last decade and the corresponding total burned forest areas in the department (Bouches-du-Rhône). More than 100 fire events occurred in 2009 and 2010 and more than 1000 ha were burned. These results combined with favoured meteorological conditions could have led to the increase of polluted episodes in Marseille. It should also be noted that 2016 and 2017 were exceptional years for the forest fire occurrence. However, $PM_{2.5}$ measurements were not measured half of the time in summer 2016 (from mid-June to beginning of August 2016) and wind conditions were not suitable for the pollution accumulation in Marseille in summer 2017. In overall, the improvement of background air pollution since the last years must also contribute to the decreased number of polluted episodes.

[Figure]

**Figure A8 – Forest fire occurrence in the "Bouche-du-Rhône" department for summer periods, from 2008 to 2018. The bars are color-coded according to the burned forest areas in ha (Data available at https://www.promethee.com/).**

- line 435: This is quite difficult to see in Figure 1.

The reference was changed to Figure 3.

- line 455: Particle number size distribution during the selected event would be helpful.

Particle number size distribution during the Christmas event is presented in Figure A9 and has now been added to the Supplement (Figure S13) with a reference in the main text.

[Figure]

**Figure A9 - Particle number size distribution (dN/dlogdP) measured by the SMPS during the Christmas event (23-24 December 2017).**

- line 470: "the aerosol chemical composition was relatively stable"

Corrected.

- line 497: Could it be possible that the polluted air masses were rather coming from the Rhone valley than bringing Pô valley polluted air masses over the Alpes mountains? Can a trajectory analysis (CWT) help to identify the potential aerosol source area?

We thank the anonymous referee for this suggestion. The CWT analysis for the nitrate (Figure A10a) clearly shows that in addition to the pollution coming from the Rhone valley, there is also a hotspot in the North-East, corresponding to the Pô Valley. Note that the Pô valley was also clearly identified as a $PM_{2.5}$ hotspot by the real-time air quality forecasting and analysis system, (Prev'air) (Figure A10b).

The CWT analysis has been added to the Figure S15.

The following sentence has been added in the section 3.4.1: *"The CWT analysis for the nitrate supports this hypothesis since it shows a potential source region in the North-East part of the Pô Valley (Figure S15)."*

[Figure]

**Figure A10 – CWT analysis for $NO_3^-$ concentrations in µg m$^{-3}$ during the long-range episodes of February 2018 at MRS-LCP (a) and $PM_{2.5}$ concentrations from the real-time air quality forecasting and analysis system "PREV'AIR" (http://www2.prevair.org/) (b).**

- line 498: Why is there a new case study event?

After major revision of the section, this case study has been removed.

- line 519: Is there any confusion here? This section aims to discuss the summer sulfate origin and figure 8 the relation between sulfate concentration and UFP over the seasons. This should rather be done earlier on the seasonal analysis part.

We agree with that remark, this element has been moved to section 3.3.2 dedicated to seasonal trends: *"While it appears that the highest UFPs mean number concentrations are observed in autumn and winter, a different picture is found when the frequency of occurrence of short term high concentration*

*episodes is investigated as a function of the season. Figure 5 shows the box plots of 15-minute average UFPs number concentrations, binned into intervals of SO$_2$ concentrations classes, a proxy industrial and shipping activity. It clearly shows that the most intense episodes occur preferentially in summer and are associated with high SO$_2$ concentrations. This season gathers more than 55% of the highest SO$_2$ concentrations (>20µg m$^{-3}$) episodes."*

- line 525: Please rephrase the sentence "during summertime, the ships traffic increases by 25 % (4319 against 3263 for the 2017-2017 period)". What are the numbers referring to?

The sentence has been modified. Numbers refer here to the sum of the ship movements (arrivals+departures) registered by the port authorities.

- line 530: I disagree a bit here since the SO2 concentration is continuously increasing from midnight to 9 o'clock, so much earlier than the ship traffic peak.

We agree with the reviewer's comment. The daily profiles of SO$_2$ concentrations in summer appear to be biased towards high values, especially between 23h00 and 04h00 UTC. These high values are observed when wind from the South-West or South-East blows to the station at this time, as revealed by the NWR polar plot and Cartesian plot of SO$_2$ concentrations (Figure A11 and Figure A12, respectively). These are rare events that do not affect significantly the mean daily profile of wind direction, because the land breeze influence largely dominates the wind pattern at night.

This has been clarified the revised version in section 3.4.2: *"It should be noted that the SO$_2$ average concentrations slightly increase earlier during the night (00h00-04h00 UTC). This is the consequence of scarce wind advections of SO$_2$ from the sea, also related to the ship traffic and not caught by the averaged wind diurnal cycle in Figure 11."*

[Figure]

**Figure A11 – NWR polar plot for SO$_2$ concentrations during summer night-time (20h00-04h00 UTC) (left) and related wind probability plot (right).**

[Figure]

**Figure A12 – NWR Cartesian plot for SO$_2$ concentrations during summertime.**

- line 556: Is the discussion on the sulfate classification needed? It was already mentioned that sulfate is fully neutralized by ammonium. No new conclusion was drawn from this sulfate fragmetns analysis. It would be interesting here to look for example, at the time series of the 3 different sulfate species and compare the MSA results with the time series of the m/z 79.

We agree with this remark. This part of the discussion and Figure 12 were removed from the manuscript. Following the reviewer suggestion, we investigated the m/z 79 time series and no significant correlation was found with any of the sulfate fragments.

- line 559: Numbering of figures 11 and 12 should be changed. Figure 12 is discussed first.

The Figure numberings were checked and corrected.

- line 574: The term long-range transport is relative here since sulfate sources look to the located relatively close to the city of Marseille.

Indeed, the term "regional" would be preferred in this case. This sentence was removed.

**Tables, Figures, and supplementary information:**

- Please used scientific notation on the axis labeling.

Scientific notations from all Figures were corrected accordingly.

-Figure 2-a: Did the authors investigate the deviation between ACSMBC and SMPS when density increase? Seems that there is a deviation for density above 1.5. Could it be linked to the presence of more sea-salt or coarse particles?

We inspected the relevant deviation in the ACSM+BC vs. SMPS comparison. These data points correspond to the beginning of December 2017. The deviation can be related to an increase of the inorganic nitrate mass fraction during this period as shown in Figure A13a (which explains the higher aerosol density). The NO$_2^+$/NO$^+$ ratio of these data points displayed typical values from ammonium nitrate (0.5-0.6) (Figure A13b)

However, there isn't any specific link between the coarse fraction (between $PM_{2.5}$ and $PM_{10}$) (Figure A13c) and the decrease in $NH_{4meas}/NH_{4pred}$ ratio (Figure A13d), limiting any statement about the presence of sea-salt particles.

[Figure]

**Figure A13 – Reconstructed $PM_1$ (ACSM + BC) vs. $PM_1$ calculated from SMPS measurements, color-coded according to $NO_{3,Inorg}$ mass fraction to $PM_1$ (a), $NO_2^+/NO^+$ ratio (b), coarse mass concentration ($PM_{10}$ - $PM_{2.5}$) (c) and $NH_{4meas}/NH_{4pred}$ ratio (d).**

Detailed inspection of the SMPS data confirmed that these periods were not associated with a significant contribution in mass of particles with size over 400 nm that would not be 100% transmitted by the lenses of the $PM_1$.

- Figure 6: Is there any reason why BCwb and BCff are presented in Fig 6a and not in Fig 6b? Wind speed would also be interesting here.

This is now presented in figure 7b.

Following the reviewer's recommendation, wind speed has been included in the Figure 7 (using a color scale for wind data).

- Figure 8: What does the number of points mean (time resolution)? Moreover, a log-scale would be helpful to better catch the number of points on each category.

The number of point refers to the number of 15-min data fitted per class. The legend was modified accordingly (referring now to *"the number of data points (15-min resolution) encountered in each bin"*) and a log scale was used.

- Figure 9: The difference between the two red colors (arrival and departure at the South terminal) is not easy to catch. Please, provide information for the wind speed intensity value.

The colors were changed and we now provide wind speed intensity on the Figure.

- Figure 10: Please include similar plots for the missing cluster 3.

This previous Figure was removed from the manuscript.

- Figure 12: Please include the expected limits of the triangle (location of organosulfates, MSA and ammonium sulfate). Ammonium sulfate from the 7/12 calibration is missing.

This figure was removed from the manuscript.

-Figure S1: The central map is rather too small and it is very difficult to distinguish the different colors on the Marseille port.

The size of the central map was increased and repositioned in the revised version.

- Figure S3 caption: Please, correct PM1 notation.

Corrected.

- Figure S5: please include the zero lines

We don't understand what the comment refers to…

 - It will make the reading of the supplementary information easier by including tables and figures directly in the corresponding text section.

This is now the case.

- Is figure S14c discussed?

A reference to this figure has been added in the supplementary text (now Figure S5).

The UFP analysis has been used in numerous studies in urban and suburban environments in Europe (del Águila et al., 2018; González et al., 2011; Hama et al., 2017b, 2017a; Reche et al., 2011; Rodríguez and Cuevas, 2007; Tobías et al., 2018). We acknowledge that results obtained from this analysis are sensitive to the sampling strategy (cut size of the particle counter, distance from the sources) and to the physicochemical properties of the primary particles (presence and size of the BC core) (Kerminen et al., 2018; Kulmala et al., 2016). We have added a mention about it in the section 3.3.2:*"It should be emphasized that results obtained from this analysis are sensitive to the sampling strategy (cut size of the particle counter, distance from the sources) and to the physicochemical properties of the primary particles (presence and size of the BC core) (Kerminen et al., 2018; Kulmala et al., 2016)."*

However, we believe that the $N_1/N_2$ estimations provided by this method over the 14 months period brings valuable information on the seasonal/daily sources contributions of UFP in Marseille. The fact that the $N_1$ and $N_2$ fraction show different diurnal patterns ($N_1$ maxima during morning and evening traffic rush hours over the year, similarly to NO concentrations; $N_2$ profile with 1-2 hours shift in autumn/winter and a broad maximum during daylight in summer coinciding with sea breeze prevalence and photochemical activity) gives us confidence in its usefulness.

In the present work, an underestimation of $N_2$ can be expected, as particles smaller than 20 nm are not included in the calculation when the TSI 3031 is used. By comparing with SMPS measurements in summer period, $N_2$ accounted for 83% of total particle number against 74% with the TSI 3031.

6) Section 3.3.1: This analysis of two pollution events do not add significantly to the publication, or generally to understanding pollution events to the site. Typically those are fairly well represented by operational atmospheric models, so the question would be how close known processes and inventories represent them, rather than relying only on in-situ aerosol measurements and local winds for this type of analysis.

The investigation of the pollution events described in Section 3.4.1 is meant to experimentally determine whether and to which extent local and/or remote pollution contribute to the $PM_{2.5}$ daily

mean threshold (25 µg m$^{-3}$) exceedance observed at the site. We now provide a more exhaustive analysis in the revised section, as the analysis was extended to all the days exceeding the PM$_{2.5}$ WHO recommendation. Based on a list of specific criteria, the exceedance days could be gathered into one of these two categories. We agree that comparison to operational atmospheric model calculations would be of great interest, but we feel this is beyond the scope of this study. Here the intention is to provide the sources and composition of the particles when such event occurs, as a support for air quality management.

Nonetheless, we hope this kind of database provides useful inputs to improve and refine the atmospheric models as uncertainties and gaps remain between simulation and measurement for submicronic aerosol (Aksoyoglu et al., 2017; Chrit et al., 2018; Lannuque et al., 2020).

7) Section 3.3.2: This section reads like a patch of several analysis hardly including more than a paragraph and a figure. The clearer example is Figure 12 and SO4 ion analysis. What is the goal with this analysis, in the context of this section and this manuscript more generally? The next two points also relate to this section.

The goal of this analysis was to investigate the chemical state of the particulate sulfur in the three different clusters. We hypothesised that more acidic sulfate particles would be brought from the harbour by the sea breeze (cluster 2). This was not confirmed by this analysis since all data points fall in the ammonium sulfate region, as shown in Figure 12. We agree that this analysis doesn't provide new insights since we had already inspected the NH4$_{pred}$/NH4$_{meas}$ ratio in Figure S4. This analysis was removed from the manuscript.

8) Cluster analyses: I'm reticent about the use of hysplit cluster on interpreting such short-scale air masses movements, particularly sea breezes. I agree that on average, the continental cluster might be more prone to sea breeze and thus be continuously fed by anthropogenic emissions, but it's a big step naming it "sea breeze", as the same can happen with Mediterranean air masses. I don't see so clearly the "discernible" differences on figure 10, which can be just due to the small statistics treated here.

We concede that the initial approach consisting in attempting to link the specific synoptic patterns with sulfate concentration using air-mass back-trajectory cluster failed in clearly separating the sulfate concentration patterns observed on the site. This was acknowledged in the text (lines 545-547: "While the clustering analysis clearly identifies the Mediterranean long range trajectories, Figure S11 shows that they still get mixed with the sea breeze when they approach the shore (as indicated by the wind sector 190°-270° characteristic of the sea breeze, and the by the sharp SO$_2$ peaks included in the Mediterranean regime periods in pink). In the revised manuscript, we have attempted to find a more appropriate way to highlight these patterns, and this was achieved using k-means clustering algorithm run on the hourly concentrations of sulfate. The idea behind this alternative approach was to classify the days into distinct group (cluster) having similar diurnal profiles and to identify these clusters as a function of wind speed, wind direction, precursor SO$_2$ concentration, N$_{10-20}$, as well as air mass origin. Despite the relatively limited dataset that constitute the summer data (≈60 days) we managed to get meaningful information on the phenomenology of sulfate in summer as three clusters are clearly identified corresponding to background, local and regional transport within the Mediterranean basin (see section 3.4.2).

9) On the cluster topic, given possible role of local sources, analysis such as figure 11 might be extremely misleading. From a quick look it seems that CWT maps follow trajectory densities. My guess would be that you'd fine similar maps for locally emitted BCff maps, for example.

The use of the CWT to investigate the potential remote sources of sulfate is motivated by its expected various origins, including regional and long range transport. Figure A14 compares the CWT plots of both sulfate and $BC_{FF}$ for the 3 clusters identified from the new analysis carried out on summer sulfate concentrations (See section 3.4.2). The fact that the hotspots associated with sulfate differ from those of black carbon (expected to reflect the local emissions), but also from the trajectory densities suggest that the CWT analysis on sulfate is robust enough to be used.

[Figure]

**Figure A14 – CWT analyses of sulfates (middle), $BC_{FF}$ (bottom) for cluster 1 (left), cluster 2 (middle) and cluster 3 (right) and corresponding trajectory densities (top).**

**Technical Check US vs UK English spelling, both are found in the text.**

L.46: replace "leave" by "live". Reference for this claim?

The text was replaced and the following reference was added:

"Rouaud, P. and Channac, Y.: Pollution de l'air par les PM10 En 2017, le seuil de l'OMS dépassé pour la moitié des résidents de la région, INSEE Analyses, INSEE, Marseille. [online] Available from: https://www.insee.fr/fr/statistiques/4250618?sommaire=4251028, 2019."

L.51: please replace "(Pandolfi et al., 2020)" by "Pandolfi et al., (2020)"

The text was replaced.

L. 72: please remove comma between "particles" and "have".

The comma was removed.

L.90: Please define LCE

LCE was defined.

L.95-96: I don't find this sentence overly clear. Does it mean it is a busy downtown area? Of course comparing with the national average this type of environment should drive the average up. Please make it clearer, and add the reference for such claim.

This was reworded as follows: "*The city also encounters the second most traffic congestion in France (TOMTOM, 2020). The number of vehicle kilometres travelled was 2.4 billion within a 5 km radius around the supersite in 2017 (AtmoSud traffic database). Considering the relative road network size it was 2.3 times higher than in the largest city in EU, London (Department for Transport, 2020).*"

L.97: replace "first" by "largest"

The text was replaced.

L.98: "berthing almost 4000 ships in 2017." Also without reference.

This number was determined from harbour data provided by the regional air quality network (AtmoSud). The following referenc*e has been added: "(based on port calls statistics registered at "Grand Port Maritime de Marseille")"*

L.102: "Driven" instead of "held"?

The text was replaced.

L.108-111: Missing reference for this sentence.

The following reference was added:

"El Haddad, I., D'Anna, B., Temime-Roussel, B., Nicolas, M., Boreave, A., Favez, O., Voisin, D., Sciare, J., George, C., Jaffrezo, J.-L., Wortham, H. and Marchand, N.: Towards a better understanding of the origins, chemical composition and aging of oxygenated organic aerosols: case study of a Mediterranean industrialized environment, Marseille, Atmospheric Chem. Phys., 13(15), 7875–7894, doi:10.5194/acp-13-7875-2013, 2013."

L.133: "The graphs on the right side display" instead of "The right graphs display".

The text was replaced.

L.123: please replace "very slight" by a percentage.

Yearly rates were added for $PM_{10}$ and $PM_{2.5}$.

L.159: "ACSM" instead of "ACMS".

The text was corrected.

L.241: "confidence" interval.

The text was corrected.

L.259-L261: I'd suggest to remove this sentence with the reference to previous work, as at this point of the manuscript there is only general description of chemical composition and no insights into their origins.

The sentence was removed.

L.260: C-ToF-AMS has not been defined.

According to the previous suggestion, the term was removed

Fig.3: The pie charts on Fig. 3 seem to suffer from low resolution. In addition, the font size indicating the seasons should be increased. The date stamp should be in the format dd-mmm-yy to avoid confusion.

The font size of the pie charts has been increased and date stamp was changed

L.264-266: Careful with the comparison, the fractional contribution of PM1-NR + BC and filter based PM2.5 are not directly comparable. The results from Putaud were from filter (wider range of species detected, especially in the refractory range such as SS and dust), and from some 15 years ago.

We assumed that $PM_1$ represented a large fraction of $PM_{2.5}$, like Marseille in our study (around 88%). SS and mineral dust being more expected in the $PM_1$ to $PM_{2.5}$ fraction, we considered it was possible to compare both data. However, we agree that the time scale between the two studies makes the comparison difficult. The sentence was deleted from the revised version.

L.266-267: Unclear why this sentence (and ref) has been added here.

After revision and rewriting of the related section, this sentence has been deleted.

L.267: "average"

The term "averaged values" was replaced by "average".

L.292: I fail to see a clear seasonality on OA concentrations, from both the plot and Table 1, so perhaps re-write this sentence.

We mentioned the seasonality effect for OA as the average concentrations in autumn/winter are 25% higher than those in spring/summer (average concentrations from Table 1). A similar trend is observed for $BC_{WB}$ as the autumn/winter concentration is 64% higher than in spring/summer.

L.302: replace "big" by "large".

The text was corrected.

L.303: remove "still".

The text was removed.

L.343-344: I find this sentence to be imprecise. It's missing reference for the link between UFP and health, and UFP can grow under the right conditions to significantly impact PM1, PM2.5 or PM10. I'd suggest to just remove it.

We agree with the referee and removed the sentence.

L.350-351: Was this seasonal variation linked to BC and OA to be observed from Fig. 3? No indication whatsoever can be seen from it.

We now refer now to figure S8 in the revised manuscript to support our remark.

L.429-431: missing reference.

The number of exceedance days is discussed from the Table 2, mentioned in the previous sentence in the manuscript.

L.435-436: rewrite sentence.

This sentence was rephrased. It reads now "The average $PM_{2.5}$ concentrations were 31.2 µg m$^{-3}$ and 36.7 µg m$^{-3}$ respectively for the local and regional pollution episodes, with average $PM_1$ concentrations of 28.7 µg m$^{-3}$ and 31.1 µg m$^{-3}$, respectively. This indicates that $PM_{2.5}$ pollution episodes are driven by $PM_1$ concentrations in both cases."

L.437-438: Unclear why Christmas event would be local pollution driver.

As stated earlier, this section was carefully rewritten in order to stress the features of these distinct events.

Fig. 6: dates for Christmas event not the same as the text.

The legend of Figure 6 (now Figure 7) has been corrected (23-24 December instead of 22-26 December).

Fig. 6: change time stamp for the format dd-mmm-yy.

Corrected.

L.514-515: repetition of information.

Corrected.

Figure S11: Correct caption N(10-20nm)

This previous Figure was removed from the manuscript.

---

## Author Response (AR2)

**General response to reviewers:**

We thank both anonymous referees #1 and #2 for providing these second reports. We have implemented the new comments in the revised version of the manuscript. The line-by-line responses to the reviewers' comments (written in black) are written in blue.

**REFEREE #1**

The authors did an impressive job in revising their manuscript and paying close attention to the comments which were very satisfactorily addressed.
I have few minor comments:

- Line 56: don't forget the work from Ovadnevaite et al., (2014) which is covering three complete years of AMS measurements. It is certainly one of the longest AMS measurements that were done.

Thanks, the reference was added.

- Lines 235-240: The authors only consider uncertainties on the ACSM. What about the loss of the most semi-volatile organic on the filter samplers?

The high correlation obtained between OA and OC ($R^2$=0.79, Figure S3) over a rather wide range of temperatures (8°C to 26.5°C), suggests that losses from semi-volatile organic compounds do not significantly affect the sample integrity. However, this potential artefact is now specified in the text. "It is possible that the chosen sampling periods for the comparison (spring and summer 2017) bias high the OM-to-OC value partly because of the volatilization of semi-volatile species from the filters. The high correlation obtained between OA and OC in Figure S3 ($R^2$=0.79), suggests that this artefact does not significantly affect the sample integrity."

- Line 246 and 247: please, include the country of the stations

The countries are now stated in the manuscript.

- Line 328: the authors only considered biogenic VOCs source of secondary organic aerosol. What about anthropogenic SOA and/or aging of anthropogenic aerosol during summertime?

In the present work the clear correlation between f44 and the temperature found in summer supports the assumption of the dominant contribution of the biogenic precursor emissions to the secondary organic aerosol formation. While it is not possible to estimate their relative contribution compared to anthropogenic precursors in the present study, we now mention the estimation of ≈70-80% provided by El Haddad et al. (2011, 2013): "This observation supports the conclusion that biogenic precursors strongly contribute to the secondary organic aerosol formation at this season. Previous studies based on radiocarbon measurements of $PM_{2.5}$ in summer evaluated that ≈70-80% of organic carbon was of non-fossil origin and was mainly attributed to biogenic secondary organic carbon (BSOC) (El Haddad et al., 2011, 2013)."

- Line 381: what about the contribution of biomass burning emission here.

Since the $N_1$ fraction was determined based on the $BC_{FF}$ contribution to UFPs, only fossil fuel emissions are expected to contribute to the increased proportion of $N_1$ observed in winter and autumn. The predominant influence of fossil fuel emissions on the increasing particle number concentrations during these seasons is supported by the increased values in the same proportions observed at daily peaks related to the traffic rush hours (7000 cm$^{-3}$ and 5900 cm$^{-3}$ in the morning, and 6000 cm$^{-3}$ and 4600 cm$^{-3}$ in the early evening, in winter and autumn, respectively, versus 5000 cm$^{-3}$ and 3500 cm$^{-3}$ in the morning and 3900 and 3500 in the evening, in spring and summer). Very recently (Casquero-Vera et al. 2021), a new approach based on the Rodríguez and Cuevas (2007) method found that biomass burning contributes in winter to 2% and 6% in the 12-25 nm and 25-100 nm size range of the UFP respectively in a urban site in Mediterranean where the biomass burning contribution to BC is comparable to the one found in Marseille (23% to the total BC vs 28% in our study). This suggests that the contribution of biomass burning emissions to UFP are expected to be low.

- Line 513: I am still not fully convinced of the importance of the nitrate mass concentration coming from the Pô valley compared to the one coming from the Rhône valley. Especially regarding the semi-volatile properties of the ammonium nitrate. Did the authors see the same results for the other chemical species (e.g. organics)?

While it is not possible to rigorously assess its contribution to the levels of ammonium nitrate observed during the PM episodes with exceedance days, we believe that the Pô valley, being one of the main pollution hotspots in Europe, is a potential source that cannot be ignored. This assumption is based on the study of the HYSPLIT air mass 72h-backtrajectories and CWT maps for $NO_3^-$ backtrajectories shown in Figure S15 and on the cold temperatures experienced when these episodes occur. We believe that cold temperatures can favour the transfer and stabilisation of nitrate in the particle phase, and thus allow its transport over long distances. Ammonium nitrate concentrations attributed to mid/long-range transport have already been observed in Europe (Bressi et al., 2014; Dall'Osto et al., 2009; Petit et al., 2015).

Similar results to those presented in Figure S15 were indeed obtained for organics, but the additional CWT analysis carried out on the $NO_3^-$/BC ratio (BC prevails for local influence from the city) and shown in Figure A1 better supports the long-range transport of ammonium nitrate.

[Figure]

Figure A1 – CWT maps for $NO_3^-$/BC during the February 2018 episode.

- Line 521: Chemical valley?

The term was corrected.

- Notation: PM1 or PM1

Corrected.

- Figure S3: some y-axis labeling is missing.

Corrected.

- Figure S12: please correct the color scale of the night-time plot.

Corrected.

references
Ovadnevaite, J., Ceburnis, D., Leinert, S., Dall'Osto, M., Canagaratna, M., O'Doherty, S., Berresheim, H., and O'Dowd, C.: Submicron NE Atlantic marine aerosol chemical composition and abundance: Seasonal trends and air mass categorization, Journal of Geophysical Research-Atmospheres, 119, 11850-11863, 10.1002/2013jd021330, 2014.

**REFEREE #2**

I thank the authors for the careful revision of the manuscript, which reads much more easily now and with a clearer message. I have just a few minor and technical comments.

Minor:
L.582: Maybe I missed it in the text, but I wonder why would cluster #1 has the highest N2 concentration? Could it be some local contamination associated with slightly different wind direction? Also, I'd suggest adding O3 diurnal variability in the clusters plot, which is a good proxy for photooxidation during those days.

We added the mean $O_3$ diurnal profiles for each cluster in Figure 12. We now refer to the ozone concentrations in the main text as following: *"The molar ratio of sulfate to total sulfur, indicative of $SO_2$-to-sulfate conversion grows when photochemical activity is expected to be at its highest (between 11h00 and 16h00 according to the highest concentrations of $O_3$ diurnal profile), from 0.3 (10h00) to 0.5 (16h00) (Figure S18)."*

About the highest $N_2$ concentration found in cluster 1: This is exactly the situation described by the referee, i.e. nearby coastal emissions of $SO_2$ and $N_{2 (10\text{-}20 \text{ nm})}$ are mixed with the regional background of sulfate once the sea breeze sets in and brings the plumes of fresh emissions to the sampling site. We state this situation in the following sentences page 18 "Cluster 1 seems to be related to the regional background of sulfate as it is associated with the lowest concentration ($\approx 2\mu g \text{ m}^{-3}$) and a flat diurnal

profile. Meanwhile, $SO_2$ and $N_{2\,(10\text{-}20\,nm)}$ increase simultaneously as the breeze sets in to peak at 4 μg m$^-$$^3$ and 8000 cm$^{-3}$ respectively at 10h00-12h00. The comparison of the daily profiles indicates that sulfate is fully decoupled from its precursors in this cluster and that no significant conversion of $SO_2$ to sulfate occurs by the time the air mass reaches the site." and later page 19 "The NWR plots confirm the hypothesis on the potential origin of sulfate and $SO_2$ in the 3 clusters: in the case of cluster 1, sulfate and $SO_2$ are disconnected to each other…"

Fig. S23: I'd suggest to move this figure to the main text, it's quite interesting and corroborates the authors interpretation of the clusters identification based on diurnal variability and back-trajectory analysis.

We moved the Figure S23 to the main text (now Figure 14).

Technical:
Title: "atmospheric dynamics"

Corrected.

Abstract: remove sentence "These episodes contribute to an increase of 6.5% of the annual PM1 concentration."

The sentence was removed.

L.65: add a comma between "years" and "aerosol mass spectrometer".

Corrected.

L.351, 353 and L.355: I think there are more line skips here than originally intended.

Corrected.

L.439: add a comma between "winter" and "N2".

Corrected.

L.468: please put the number "2" in writing.

Corrected.

Fig. 7: the axis labels are quite close, making it difficult to read it. I suggest to favor skipping line whenever possible (PBL /n (m), for example) and perhaps increase the figure height.

The axis labels were adjusted accordingly.

L.510: I'd prefer to replace "secondary" with "processed", as f44 is not a unique tracer of the former.

Corrected.

L.532: please add "fresh" before biomass burning.

Corrected.

L.620: Define concentration of which parameter (NR-PM1, PM1, PM2.5)

We now refer to the "$PM_1$ concentration level".

**References:**

Bressi, M., Sciare, J., Ghersi, V., Mihalopoulos, N., Petit, J.-E., Nicolas, J. B., Moukhtar, S., Rosso, A., Féron, A., Bonnaire, N., Poulakis, E., and Theodosi, C.: Sources and geographical origins of fine aerosols in Paris (France), Atmos. Chem. Phys., 14, 8813–8839, https://doi.org/10.5194/acp-14-8813-2014, 2014.

Casquero-Vera, J. A., Lyamani, H., Titos, G., Minguillón, M. C., Dada, L., Alastuey, A., Querol, X., Petäjä, T., Olmo, F. J., and Alados-Arboledas, L.: Quantifying traffic, biomass burning and secondary source contributions to atmospheric particle number concentrations at urban and suburban sites, Science of The Total Environment, 768, 145282, https://doi.org/10.1016/j.scitotenv.2021.145282, 2021.

Dall'Osto, M., Harrison, R. M., Coe, H., Williams, P. I., and Allan, J. D.: Real time chemical characterization of local and regional nitrate aerosols, Atmos. Chem. Phys., 12, 2009.

El Haddad, I., Marchand, N., Temime-Roussel, B., Wortham, H., Piot, C., Besombes, J.-L., Baduel, C., Voisin, D., Armengaud, A., and Jaffrezo, J.-L.: Insights into the secondary fraction of the organic aerosol in a Mediterranean urban area: Marseille, 11, 2059–2079, https://doi.org/10.5194/acp-11-2059-2011, 2011.

El Haddad, I., D'Anna, B., Temime-Roussel, B., Nicolas, M., Boreave, A., Favez, O., Voisin, D., Sciare, J., George, C., Jaffrezo, J.-L., Wortham, H., and Marchand, N.: Towards a better understanding of the origins, chemical composition and aging of oxygenated organic aerosols: case study of a Mediterranean industrialized environment, Marseille, 13, 7875–7894, https://doi.org/10.5194/acp-13-7875-2013, 2013.

Petit, J.-E., Favez, O., Sciare, J., Crenn, V., Sarda-Estève, R., Bonnaire, N., Močnik, G., Dupont, J.-C., Haeffelin, M., and Leoz-Garziandia, E.: Two years of near real-time chemical composition of submicron aerosols in the region of Paris using an Aerosol Chemical Speciation Monitor (ACSM) and a multi-wavelength Aethalometer, 15, 2985–3005, https://doi.org/10.5194/acp-15-2985-2015, 2015.

Rodríguez, S. and Cuevas, E.: The contributions of "minimum primary emissions" and "new particle formation enhancements" to the particle number concentration in urban air, 38, 1207–1219, https://doi.org/10.1016/j.jaerosci.2007.09.001, 2007.